# Latent mixed-effect models for high-dimensional longitudinal data

**Priscilla Ong**                                                   *priscilla.ong@aya.yale.edu*
*Department of Computer Science*
*Aalto University*

**Manuel Haußmann**\*                                                 *haussmann@imada.sdu.dk*
*Department of Mathematics and Computer Science*
*University of Southern Denmark*

**Otto Lönnroth**                                                      *otto.lonnroth@aalto.fi*
*Department of Computer Science*
*Aalto University*

**Harri Lähdesmäki**                                                *harri.lahdesmaki@aalto.fi*
*Department of Computer Science*
*Aalto University*

**Reviewed on OpenReview:** *https://openreview.net/forum?id=7A96yteeF9*

## Abstract

Modelling longitudinal data is an important yet challenging task. These datasets can be high-dimensional, contain non-linear effects and feature time-varying covariates. Gaussian process (GP) prior-based variational autoencoders (VAEs) have emerged as a promising approach due to their ability to model time-series data. However, they are costly to train and struggle to fully exploit the rich covariates characteristic of longitudinal data, making them difficult for practitioners to use effectively. In this work, we leverage linear mixed models (LMMs) and amortized variational inference to provide conditional priors for VAEs, and propose LMM-VAE, a scalable, interpretable and identifiable model. We highlight theoretical connections between it and GP-based techniques, providing a unified framework for this class of methods. Our proposal performs competitively compared to existing approaches across simulated and real-world datasets.

## 1 Introduction

Longitudinal datasets, typically containing repeated measurements of individuals over time, have applications in numerous fields such as education, psychology, the social sciences, and the biomedical field. Longitudinal study designs are particularly useful for revealing associations between explanatory covariates and a response variable, such as the relationship between risk factors and disease progression (Diggle, 2002), but require appropriate statistical tools that can account for correlations both within subjects as well as across subjects. Analysis of univariate or low-dimensional longitudinal data is dominated by various linear mixed models (LMMs), also known as multi-level models, along with other additive modelling approaches. However, existing methods scale poorly to relevant longitudinal datasets, such as electronic health records, which are often high-dimensional (Zipunnikov et al., 2014) and contain non-linear effects, time-varying covariates, and missing values (Ramchandran et al., 2021).

Variational autoencoders (VAEs) are a class of models commonly used for representation learning and generative modelling (Rezende et al., 2014; Kingma & Welling, 2014). Nevertheless, they cannot be directly

---

\*Work done at the Department of Computer Science, Aalto University.

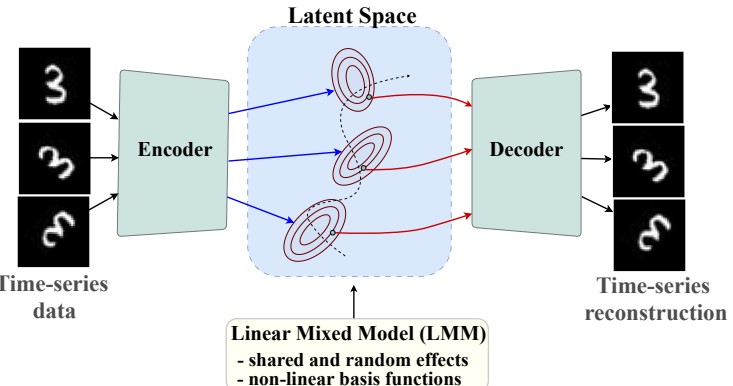

Figure 1: *Overview of LMM-VAE.* The latent space is modulated by auxiliary covariates parametrized by a linear mixed model (LMM).

applied to longitudinal data as they assume that observations are independent and identically distributed (i.i.d), thereby failing to capture correlations between samples. This limitation has been partially addressed by recently proposed Gaussian process (GP) prior VAEs that use auxiliary covariates to impose correlation structure on the samples' latent representations. However, such models have cubical complexity with respect to (w.r.t) the number of samples, necessitating model simplifications (Fortuin et al., 2020) or specific GP approximations to be applied as part of the VAE model fitting (Casale et al., 2018; Ramchandran et al., 2021). Another challenge concerns fully exploiting the rich auxiliary covariates available for modelling. While conditional VAEs (CVAEs) (Sohn et al., 2015) can easily incorporate any number of covariates, limited work has tackled the problem of finding an appropriate time-series or longitudinal model that can effectively scale to include large number of covariates into the VAE prior. Given the potential upside in model performance, addressing the model's scalability, specifically to include more covariates, is a fruitful avenue of research.

Chosing a specific parametrization for the prior is a *model selection problem*. Besides using a sufficiently rich model class, including appropriate covariates is integral (George, 2000) for modelling the latent space effectively. Different from previous work, we propose modelling the prior using a linear mixed model with appropriately chosen basis functions. While simple, this model class is scalable, vastly simplifies the training procedure within standard deep learning frameworks, and enjoys several advantages discussed below.

**Contributions.** We propose the *Linear Mixed Model VAE (LMM-VAE)*, which is capable of handling high-dimensional data and large dataset sizes, modelling an arbitrary number of covariates in the prior, can be adapted to problems of different complexity via basis functions, and enjoys the advantages of being *interpretable* and *identifiable* by way of parametrization. We demonstrate that LMM-VAEs are competitive against commonly used GP-based methods for integrating auxiliary covariates into the prior. We also highlight theoretical connections between our proposed model and GP prior VAEs, bridging the gap between our work and existing methods. From both a practical and theoretical standpoint, LMMs show promise as an alternative VAE prior, especially in the presence of high dimensional covariates. Section 2 compares previous methods with our proposed LMM-VAE. See Figure 1 for a visual summary.

## 2 Related Work

The literature on extending VAEs' modelling capacity is vast. It spans from extending the expressivity of variational posterior distributions, e.g., using a multivariate Gaussian with a tridiagonal inverse covariance structure (Fortuin et al., 2020) or normalizing flows (Rezende & Mohamed, 2015), to enabling more informative priors compared to the standard normal used in plain VAEs (Kingma & Welling, 2019). Alternative research efforts focus on endowing the model with desirable characteristics, such as a disentangled latent space (Higgins et al., 2016; Zhao et al., 2019). Our work enhances the latent space representation by incorporating auxiliary covariates into the prior, such as time-varying side-profile information.

Table 1: Comparison of related methods.

| Model | Prior | Covariates | Minibatching | Identifiable | References |
|---|---|---|---|---|---|
| CVAE | (I.I.D)[1]Gaussian | arbitrary | ✓ | ✗/ ✓[1] | Sohn et al. (2015) |
| GPP-VAE | GP | limited | pseudo | unknown | Casale et al. (2018) |
| SVGP-VAE | GP | limited | ✓ | unknown | Jazbec et al. (2021) |
| LVAE | GP | arbitrary | ✓ | unknown | Ramchandran et al. (2021) |
| Longitudinal VAE | LM | limited | ✓ | unknown | Sauty & Durrleman (2022) |
| LMM-VAE | LMM (or GP) | arbitrary | ✓ | ✓ | Our work |

**Structured Variational Autoencoder (SVAE).** We build upon the ideas of Johnson et al. (2016), who combine neural networks and probabilistic graphical models to provide structured latent representations. LMM-VAE's enhancements are designed to induce structure using arbitrarily many covariates characteristic of longitudinal data. Incorporating such structure is important in many real-world applications, e.g., Sauty & Durrleman (2022) characterize the progression of Alzheimer's disease via a mixed-effect longitudinal model in the VAE setting. However, their model's parametric structure is confined to disease progression and requires an internal procedure of Markov chain Monte Carlo sampling. LMM-VAE, in contrast, provides a generalized rendition of this approach.

**Gaussian Process Variational Autoencoder.** The family of GP prior VAEs is most relevant as related work in modelling the prior. Within the GP-VAE framework, there have been multiple developments (Casale et al., 2018; Ashman et al., 2020; Ramchandran et al., 2021; Jazbec et al., 2021; Zhu et al., 2023), where the GP's expressiveness and smoothness are leveraged to enable flexible, yet robust VAE priors. We focus on those GP-VAEs that are most compatible with modelling longitudinal data, i.e. repeated measurements with auxiliary covariates, including individual-specific information.

While GP-based priors (Casale et al., 2018; Jazbec et al., 2021; Ramchandran et al., 2021) can model longitudinal data, using a GP model component comes with a host of difficulties. *Firstly*, learning scales cubically with respect to the number of samples (Rasmussen & Williams, 2006), which limits the model's scalability. Efforts to reduce training complexity include approximating the GP priors via Taylor approximation (Casale et al., 2018) or through inducing points (Ramchandran et al., 2021; Jazbec et al., 2021). For the latter, optimizing the inducing point locations is challenging (Bauer et al., 2016) and may complicate training due to the coupled learning of both the latent variables and the inducing points (Titsias, 2009; Hensman et al., 2013). Categorical inputs, which are typically modelled in longitudinal set-ups, may exacerbate this issue as they are not amenable to gradient-based optimization. In addition, unless appropriately designed and implemented, these approximations may result in diminished expressiveness of the GP-priors.

*Secondly*, modelling all available covariates with a GP is non-trivial. Jazbec et al. (2021)'s approach may be ill-suited for this task as it assumes low-dimensional covariates. Casale et al. (2018) posit that the auxiliary information can be represented by an object and a view kernel, where the dataset comprises of objects in different views. However, constructing these kernels from high dimensional side-profile information consisting of continuous and categorical covariates remains unclear. Meanwhile, Ramchandran et al. (2021) propose using additive kernels including all covariates, with each component or pair implementing a specific kernel. However, performance gains may be limited by the challenges of training the GPs as a VAE prior.

**Linear Models.** For certain modelling tasks, linear models (LM) (Galton, 1886; Fisher, 1925) are a practical and competitive alternative to highly parametrized models. For low-dimensional data, LMs and LMMs have become the standard tool of analyzing various biomedical and longitudinal data (Mundo et al., 2022). LMs are also applied in the context of high-dimensional data, such as when applied to structural and functional brain imaging data (Schulz et al., 2020) or when performing dimensionality reduction using principal component analysis (PCA) or its generative counterpart, probabilistic PCA (PPCA).

However, the assumption of a strictly linear relationship between covariates and response variables may be overly restrictive in certain contexts. Instead of enforcing linearity throughout the model, we impose this constraint only within the latent space while allowing for more complex, nonlinear mappings between the

latent space and the response variables. Moreover, we relax the strict linearity in latent space via the basis function expansion. By combining the advantages of LMMs with the flexibility of neural networks, LMM-VAE extends the traditional LMM framework, providing a more powerful and adaptable approach for modelling complex data.

## 3 Background

**Linear Models.** Consider a pair $(\boldsymbol{x}, \boldsymbol{z})$, where $\boldsymbol{x} = (x_1, \ldots, x_Q)^T \in \mathcal{X} = \prod_{i=1}^{Q} \mathcal{X}_i$ is $Q$-dimensional covariate vector and $\boldsymbol{z} \in \mathcal{Z} = \mathbb{R}^L$ is $L$-dimensional response variable. The *linear model (LM)* for $(\boldsymbol{x}, \boldsymbol{z})$ is

$$\boldsymbol{z} = \boldsymbol{a}_1 x_1 + \cdots + \boldsymbol{a}_Q x_Q + \boldsymbol{\epsilon} = A\boldsymbol{x} + \boldsymbol{\epsilon}, \tag{1}$$

where $\boldsymbol{a}_i \in \mathbb{R}^L$, $A = (\boldsymbol{a}_1, \ldots, \boldsymbol{a}_Q) \in \mathbb{R}^{L \times Q}$, and $\boldsymbol{\epsilon} \sim N(\mathbf{0}, \sigma_z^2 I)$ assuming equal variance. For $N$ pairs of covariates and response variables, $\{(\boldsymbol{x}_n, \boldsymbol{z}_n)\}_{n=1}^{N}$, the linear model is given as

$$Z = AX + E,$$

where $Z = (\boldsymbol{z}_1, \ldots, \boldsymbol{z}_n)$, $X = (\boldsymbol{x}_1, \ldots, \boldsymbol{x}_n)$ and $E = (\boldsymbol{\epsilon}_1, \ldots, \boldsymbol{\epsilon}_N)$ such that

$$p(Z|X) \triangleq \prod_{n=1}^{N} \mathcal{N}(\boldsymbol{z}_n | A\boldsymbol{x}_n, \sigma_z^2 I). \tag{2}$$

Without loss of generality (w.l.o.g.), we assume that each covariate is either continuous or binary since categorical covariates can be one-hot encoded into binary values. To model non-linear effects in $\mathcal{Z}$, we follow common practice and extend the linear model with non-linear basis functions $\phi(\cdot)$ for continuous covariates, $x' = \phi(x)$, such as $x' = x^2$, $x' = \sin(x)$, $x' = \cos(x)$, etc. We elaborate on this in Section 6.

Longitudinal datasets consist of repeated measurements of instances (e.g. patients) over time and are commonly modelled using *linear mixed models (LMM)* (Laird & Ware, 1982).While standard LMs are designed to model effects that are shared across all instances (fixed effects), LMMs can simultaneously model both fixed and random effects. Random effects represent individual- or group-specific variations, accounting for differences across individuals or groups, whereas fixed effects have a consistent influence across all observations. Modelling both types of effects captures individual- and group-level variations while maintaining generalizability across observations, resulting in more structured and meaningful latent space representations. W.l.o.g., we assume that the covariates are ordered as

$$\boldsymbol{x} = (\underbrace{x_1, \ldots, x_S}_{\boldsymbol{x}_S^T}, \underbrace{x_{S+1}, \ldots, x_{S+R}}_{\boldsymbol{x}_R^T})^T,$$

where $Q = S + R$, with the first $S$ covariates modelling shared effects, while the remaining $R$ model random effects. For example, $\boldsymbol{x}_S$ may include the age or gender of an individual that we would like to model as a shared effect. Similarly, $\boldsymbol{x}_R$ may include binary covariates corresponding to the identity of all instances (patients), which can be used to model e.g. instance-specific random offset terms. Covariates $\boldsymbol{x}_R$ may also include other types of variables, such as interaction terms specifying random effects for arbitrary subgroups of individuals. We write the LMM as

$$\boldsymbol{z} = \underbrace{(\boldsymbol{a}_1, \ldots, \boldsymbol{a}_S)}_{A_S} \boldsymbol{x}_S + \underbrace{(\boldsymbol{a}_{S+1}, \ldots, \boldsymbol{a}_{S+R})}_{A_R} \boldsymbol{x}_R + \boldsymbol{\epsilon}$$

$$= \underbrace{A_S \boldsymbol{x}_S}_{\text{shared effects}} + \underbrace{A_R \boldsymbol{x}_R}_{\text{random effects}} + \boldsymbol{\epsilon} = \underbrace{(A_S, A_R)}_{A} \boldsymbol{x} + \boldsymbol{\epsilon}.$$

Thus, we use the same compact notation shown in Eqs. equation 1 and equation 2 for both LMs and LMMs.

Thus far, our current formulation does not fully align with conventions commonly used in the LMM literature. To improve clarity and align more closely with conventions in the LMM literature, we include a simple example in Appendix A that formalizes the problem in the context of LMM-VAE.

---

[1]Identifiability depends on the CVAE parametrization.

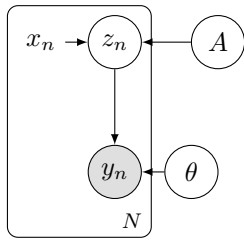

Figure 2: *LMM-VAE.* A plate diagram with probabilistic priors on all parameters. Shaded and blank circles refer to observed and latent variables, respectively. For a more detailed illustration of both fixed and random effects, refer to Appendix B.

**Variational Autoencoders.** We assume observations $\boldsymbol{y} \in \mathcal{Y} = \mathbb{R}^D$, and latent variables $\boldsymbol{z} \in \mathcal{Z} = \mathbb{R}^L$, where $L \ll D$. Given a dataset $Y = (\boldsymbol{y}_1, \ldots, \boldsymbol{y}_N)$ containing $N$ observations, we assume that $Y$ is generated by latent variables $Z = (\boldsymbol{z}_1, \ldots, \boldsymbol{z}_N)$ and write the joint generative model for a single observation as $p_\omega(\boldsymbol{y}, \boldsymbol{z}) = p_\theta(\boldsymbol{y}|\boldsymbol{z})p_\varphi(\boldsymbol{z})$, where $\omega = \{\theta, \varphi\}$. For vanilla latent variable models, $\boldsymbol{z}$ is typically assumed to have the standard normal prior, i.e., $\boldsymbol{z} \sim \mathcal{N}(\boldsymbol{0}, I)$, where $I$ is the $L$-by-$L$ identity matrix. We are typically interested in inferring the posterior $p_\omega(\boldsymbol{z}|\boldsymbol{y})$, which is generally intractable due to the need for computing the evidence $p(\boldsymbol{y}) = \int p_\theta(\boldsymbol{y}|\boldsymbol{z})p_\varphi(\boldsymbol{z})d\boldsymbol{z}$.

VAEs (Rezende & Mohamed, 2015; Kingma & Welling, 2014) rely on amortized variational inference, by specifying a parameterized approximation $q_\psi(\boldsymbol{z}|\boldsymbol{y})$, known as the *encoder*, to the intractable true posterior $p_\omega(\boldsymbol{z}|\boldsymbol{y})$. Its parameters $\phi$ are optimized jointly with $\omega$ by maximizing an *evidence lower bound (ELBO)*

$$\log p_\omega(Y) \geq \mathbb{E}_{q_\psi}\left[\log p_\theta(Y|Z)\right] - \text{KL}\left(q_\psi(Z|Y) \parallel p_\varphi(Z)\right),$$

where KL denotes the Kullback-Leibler divergence. Conditional VAEs (CVAEs) (Sohn et al., 2015) additionally condition the model on additional side information, i.e., auxiliary covariates, $\boldsymbol{x} \in \mathcal{X}$. CVAEs typically assume either $p_\omega(\boldsymbol{y}, \boldsymbol{z}|\boldsymbol{x}) = p_\theta(\boldsymbol{y}|\boldsymbol{z}, \boldsymbol{x})p_\varphi(\boldsymbol{z})$, or $p_\omega(\boldsymbol{y}, \boldsymbol{z}|\boldsymbol{x}) = p_\theta(\boldsymbol{y}|\boldsymbol{z}, \boldsymbol{x})p_\varphi(\boldsymbol{z}|\boldsymbol{x})$. The encoder of a CVAE can be extended similarly by augmenting by the covariates, $q_\psi(\boldsymbol{z}|\boldsymbol{y}, \boldsymbol{x})$. Their ELBO objective is obtained by conditioning the probabilities of the generative model and the encoder with covariates $X = (\boldsymbol{x}_1, \ldots, \boldsymbol{x}_N)$. GP prior VAEs also belong to the family of CVAEs and assume a joint prior $p_\omega(\boldsymbol{y}, \boldsymbol{z}|\boldsymbol{x}) = p_\theta(\boldsymbol{y}|\boldsymbol{z})p_\varphi(\boldsymbol{z}|\boldsymbol{x})$ where $p_\varphi(\boldsymbol{z}|\boldsymbol{x})$ is a GP.

## 4 Linear Mixed Model VAEs

We propose a structured VAE by incorporating LMMs into the prior of the VAE, and propose two variants of the model, which we collectively refer to as LMM-VAE. This model can accommodate an arbitrary number of auxiliary covariates in the prior, model both shared and random effects, and allow efficient model learning via the global parametrization.

### 4.1 Ordinary LMM-VAE

The first model variant, which we call ordinary LMM-VAE (oLMM-VAE), assumes that a standard LMM is used to model the latent variable. Assuming a prior on matrix $A$ and an optionally probabilistic decoder, we formulate the generative model for a sample $\boldsymbol{y}$ with covariates $\boldsymbol{x}$ as

$$\theta \sim p(\theta) \tag{3}$$
$$A \sim p(A) \tag{4}$$
$$\boldsymbol{z}|A, \boldsymbol{x} \sim \mathcal{N}(\boldsymbol{z}|A\boldsymbol{x}, \sigma_z^2 I) \tag{5}$$
$$\boldsymbol{y}|\boldsymbol{z}, \theta \sim p(\boldsymbol{y}|\boldsymbol{z}, \theta), \tag{6}$$

where $\theta$ parameterizes a decoder $f_\theta : \mathcal{Z} \to \mathcal{Y}$ as well as the Gaussian (or other) likelihood model. If needed, we can define different priors for shared and random effects, $p(A) = p(A_S)p(A_R)$.

We assume a factorizing variational posterior $q_\phi(A, \theta, Z|Y) = \prod_n q_\phi(\boldsymbol{z}_n|\boldsymbol{y}_n)q(A)q(\theta)$, where all approximate posteriors are Gaussian, and $\phi$ denotes an encoder network that maps an individual observation $\boldsymbol{y}_n$ into parameters of the Gaussians, mean $\boldsymbol{\mu}_n$ and diagonal covariance $\boldsymbol{\sigma}_n^2$. The ELBO is given as

$$\log p(Y|X) \geq \sum_n \Big( \mathbb{E}_{q(\theta)q_\psi(\boldsymbol{z}_n|\boldsymbol{y}_n)} \left[ \log p(\boldsymbol{y}_n|\boldsymbol{z}_n, \theta) \right]$$

$$- \mathbb{E}_{q(A)} \left[ \mathrm{KL} \left( q_\psi(\boldsymbol{z}_n|\boldsymbol{y}_n) \parallel p(\boldsymbol{z}_n|A, \boldsymbol{x}_n) \right) \right] \Big)$$

$$- \mathrm{KL} \left( q(A) \parallel p(A) \right) - \mathrm{KL} \left( q(\theta) \parallel p(\theta) \right).$$

It is straightforward to optimize this ELBO with mini-batch-based stochastic gradient descent (SGD) as the parametrization of LMM-VAE model is global.

The first expectation can be approximated via Monte Carlo sampling, while the remaining terms are analytically tractable assuming Gaussian priors.

For the $n$-th observation, we write the LMM prior mean for an arbitrary dimension $l$ of $\boldsymbol{z}$ as $z_l = \overline{\boldsymbol{a}}_l\boldsymbol{x} + \epsilon_l$, where $\overline{\boldsymbol{a}}_l$ denotes the $l$-th row of $A$. Using the independence of the latent dimensions in both prior and posterior, the second term of the ELBO becomes

$$\mathbb{E}_{q(A)} \left[ \mathrm{KL} \left( q_\psi(\boldsymbol{z}|\boldsymbol{y}) \parallel p(\boldsymbol{z}|A, \boldsymbol{x}) \right) \right]$$

$$= \sum_l \mathbb{E}_{q(A)} \left[ \mathrm{KL} \left( q_\psi(z_l|\boldsymbol{y}) \parallel p(z_l|\overline{\boldsymbol{a}}_l, \boldsymbol{x}) \right) \right]$$

$$= \sum_l \mathbb{E}_{q(A)} \left[ \frac{(\mu_l - \overline{\boldsymbol{a}}_l\boldsymbol{x})^2}{2\sigma_z^2} \right] + \frac{1}{2} \left( \frac{\sigma_l^2}{\sigma_z^2} - 1 - \log \frac{\sigma_l^2}{\sigma_z^2} \right),$$

where $\boldsymbol{\mu} = (\mu_1, \ldots, \mu_L)^T$ and $\boldsymbol{\sigma}^2 = (\sigma_1^2, \ldots, \sigma_L^2)^T$ are the parameters of $q_\psi(\boldsymbol{z}|\boldsymbol{y})$. We simplify the numerator of the first term further

$$\mathbb{E} \left[ (\mu_l - \overline{\boldsymbol{a}}_l\boldsymbol{x})^2 \right] = (\mu_l - \mathbb{E}\left[\overline{\boldsymbol{a}}_l\right]^T \boldsymbol{x})^2 + \mathrm{var}(\overline{\boldsymbol{a}}_l)^T(\boldsymbol{x} \circ \boldsymbol{x}),$$

where $\circ$ is the Hadamard product, and the expectations with respect to $q(A)$ are analytically tractable.

We could alternatively treat the generative model parameters $\theta$ and $A$ as deterministic hyperparameters, in which case the LMM-VAE model reduces to Equations (5) and (6). The reconstruction and regularization terms in the ELBO then simplify to $\mathbb{E}_{q_\psi(\boldsymbol{z}_n|\boldsymbol{y}_n)} \left[ \log p(\boldsymbol{y}_n|\boldsymbol{z}_n, \theta) \right]$ and $\mathrm{KL} \left( q_\psi(\boldsymbol{z}_n|\boldsymbol{y}_n) \parallel p(\boldsymbol{z}_n|A, \boldsymbol{x}_n) \right)$, respectively.

## 4.2 Structured LMM-VAE

In structured LMM-VAE (sLMM-VAE), we impose further structure and inductive bias into LMM-VAE for datasets that do not require observation specific variation in the latent space. Instead of having $\boldsymbol{\epsilon}_n \sim \mathcal{N}(\boldsymbol{0}, \sigma_z^2 I)$ for each $n \in \{1, \ldots, N\}$, we assume that variation in the latent space is shared for observations arising from the same group. Suppose that the $N$ observations are divided into $K$ groups. Throughout this work, we assume that each group consists of observations from a single instance (i.e., a patient), but groups could be defined more generally. W.l.o.g., we assume that the $N$ observations are ordered such that the first $n_1$ observations, $I_1 = \{1, \ldots, n_1\}$, belong to the first instance, the next $n_2$ observations, $I_2 = \{n_1 + 1, \ldots, n_1 + n_2\}$, to the second instance, and so on. We define the generative model for $K$ groups consisting of $\sum_{k=1}^{K} n_k = N$ observations, as

$$\theta \sim p(\theta),$$
$$A \sim p(A),$$
$$\boldsymbol{\epsilon}^{(k)} \sim \mathcal{N}(\boldsymbol{0}, \sigma_z^2 I), \text{ for } k = 1, \ldots, K,$$
$$Z|A, X = AX + E', \tag{7}$$
$$Y|Z, \theta \sim \prod_n p(\boldsymbol{y}_n|\boldsymbol{z}_n, \theta), \tag{8}$$

where

$$E' = (\underbrace{\boldsymbol{\epsilon}^{(1)}, \ldots, \boldsymbol{\epsilon}^{(1)}}_{n_1 \text{ times}}, \ldots, \underbrace{\boldsymbol{\epsilon}^{(K)}, \ldots, \boldsymbol{\epsilon}^{(K)}}_{n_K \text{ times}})$$

$$= (E_1, \ldots, E_K) \in \mathbb{R}^{L \times N}.$$

Note that Eq. equation 7 specifies that all samples from the $k$-th group share the same latent variation $\boldsymbol{\epsilon}^{(k)}$, i.e. $\boldsymbol{z}_n = A\boldsymbol{x}_n + \boldsymbol{\epsilon}^{(k)}$ for all $n \in I_k$. The encoder for the amortized variational inference needs to be designed such that the shared latent space variation is accounted for.

### 4.3 sLMM-VAE encoder architecture

Let $\psi$ denote the encoder network for the oLMM-VAE model from Section 4.1 that maps an individual observation $\boldsymbol{y}_n$ into parameters of the variational approximation, $\boldsymbol{\mu}_n$ and $\boldsymbol{\sigma}_n^2$. By allowing the generative model to share the linear model parameters $A$ with the encoder, we define amortized variational parameters $\boldsymbol{\mu}^{(k)}$ and $\boldsymbol{\sigma}^{(k)2}$ for Gaussian approximation $q_\phi(\boldsymbol{\epsilon}^{(k)}|Y, A, X)$ as follows:

$$\boldsymbol{\mu}^{(k)} = \frac{1}{n_k} \sum_{n \in I_k} A\boldsymbol{x}_n - \boldsymbol{\mu}_n,$$

$$\boldsymbol{\sigma}^{(k)2} = \frac{1}{n_k - 1} \sum_{n \in I_k} \left( \boldsymbol{\mu}^{(k)} - (A\boldsymbol{x}_n - \boldsymbol{\mu}_n) \right)^2, \tag{10}$$

where $\boldsymbol{\mu}_n$ is obtained from $\boldsymbol{y}_n$ with $\psi$, and $(\cdot)^2$ is computed element-wise. By this formulation, all observations belonging to the $k^{th}$ group would share the same amortized variational parameters, thereby enforcing more structured representations at the individual level.

### 4.4 Predictive training for LMM-VAE

As a last variation, we present a model-agnostic training routine that could yield improved predictions for *conditional generation*. First introduced by Sohn et al. (2015), the *Gaussian stochastic neural network (GSNN)* relies on training a predictive model, where the variational approximation is set to be identical to the prior, i.e. $q_\psi(\boldsymbol{z}|\boldsymbol{x}, \boldsymbol{y}) = p(\boldsymbol{z}|\boldsymbol{x})$. The prior is then directly trained for the task of data generation, and the ELBO loss for the GSNN model is given as

$$\mathcal{L}_{\text{GSNN}}(Y|X, A, \theta) = \frac{1}{S} \sum_{s=1}^{S} \sum_{n=1}^{N} \log p(\boldsymbol{y}_n | \boldsymbol{z}_{n,s}, \theta), \tag{11}$$

$$\text{where} \quad \boldsymbol{z}_{n,s} = A\boldsymbol{x}_n + \boldsymbol{\epsilon}_{n,s}, \tag{12}$$

$$\boldsymbol{\epsilon}_{n,s} \sim \mathcal{N}(\boldsymbol{0}, \sigma_z^2 I), \tag{13}$$

where we use $S$ Monte Carlo samples to estimate the log-likelihood. GSNN-based training objective for the sLMM-VAE model variant can be defined similarly. When the auxiliary covariates are sufficiently informative, using GSNN-based training could yield competitive performance for data generation. See Appendix D for further discussion of the GSNN loss.

## 5 Identifiability

Drawing on Khemakhem et al. (2020)'s work, LMM-VAE's design as a conditional latent variable model guarantees it the property of *identifiability*. Khemakhem et al. (2020) require the following structure

$$p_f(\boldsymbol{y}|\boldsymbol{z})p_{\boldsymbol{T}, \lambda}(\boldsymbol{z}|\boldsymbol{x}),$$

where $p_{\boldsymbol{T}, \lambda}(\boldsymbol{z}|x)$ belongs to the exponential family,

$$p(\boldsymbol{z}|\boldsymbol{x}) \propto \prod_{i=1}^{L} m_i(z_i) \exp\left[ \sum_{j=1}^{k} T_{i,j}(z_i)\lambda_{i,j}(\boldsymbol{x}) \right], \tag{14}$$

with $m$ being the base measure, $\boldsymbol{T}$ the sufficient statistics, and $\lambda(\boldsymbol{x})$ the corresponding parameters. $p(\boldsymbol{y}|\boldsymbol{z})$ needs to be decomposable into $\boldsymbol{y} = f(\boldsymbol{z}) + \epsilon$ with $\epsilon \sim p(\epsilon)$, and $L < D$. Additionally, we require $\boldsymbol{x}$ to be sufficiently diverse, such that it consists of at least $kL + 1$ distinct values, where $k$ is the number of sufficient statistics, an injective function $f$, as well as additional technical conditions summarized in Appendix G. Then, the model is identifiable in $(f, \boldsymbol{T}, \boldsymbol{\lambda})$ up to rotations and translations. Given our choice of priors in equation 5 and equation 7, and assuming an injective decoder, both our oLMM-VAE as well as our sLMM-VAE variation fulfill the constraints. See Appendix G for a detailed discussion and results on the Project Data Sphere dataset.

**Motivation.** Identifiability ensures that the true joint distribution over observed and latent variables, up to rotations and translations, can be estimated (Khemakhem et al., 2020). In other words, identifiability implies consistency of parameter estimation, up to certain transformations. Although an asymptotic theoretical guarantee, identifiability is a desirable property because it enables a meaningful interpretation of model coefficients and the data generation mechanism. In real-world applications utilizing longitudinal datasets, downstream analyses following model fitting would allow practitioners to gather insights and understand how model predictions are generated, based on the estimated parameters. These properties are useful because many real-world applications of longitudinal data seek to address questions that extend beyond mere model fitting.

## 6 GP prior VAEs as LMM-VAEs

As noted in Section 3, LMs and LMMs can be made arbitrarily complex with additional basis functions, while maintaining linearity with respect to the parameters. This basis function extension highlights a useful connection between LMM-VAEs and GP prior VAEs. This connection builds upon a well-known result that GPs correspond to Bayesian linear regression with infinitely many basis functions (see e.g. (Rasmussen & Williams, 2006)). Here, we use the spectral domain representation, but similar constructions could be derived for other basis, such as eigenfunctions from the Mercer's theorem (Rasmussen & Williams, 2006) or Laplace eigenfunctions (Solin & Särkkä, 2019).

Based on the spectral domain representation, a univariate stationary covariance function $k(r)$, where $r = x - x'$, can be approximated using a finite basis function expansion (Rasmussen & Williams, 2006)

$$k(r) = \frac{1}{2\pi} \int_{-\infty}^{\infty} s(\omega)e^{i\omega r} d\omega \approx \frac{\sigma^2}{M} \sum_{m=1}^{M} \cos(\omega_m r) = \tilde{k}(r),$$

where $s(\omega)$ is the kernel's spectral density at frequency $\omega$, $i$ the imaginary number, $\sigma^2$ the kernel variance, $M$ the number of terms in the approximation, and $\omega_m$ are either regular or random Fourier frequencies (with a distribution proportional to $s(\omega)$). To construct the basis functions, we use the trigonometric identity $\cos(u - v) = \cos(u)\cos(v) + \sin(u)\sin(v)$ (Tompkins & Ramos, 2018), and represent $\tilde{k}(r)$ using the feature map

$$\phi(x) = (\cos(\omega_1 x), \ldots, \cos(\omega_M x),$$
$$\sin(\omega_1 x), \ldots, \sin(\omega_M x))^T.$$

Following Hensman et al. (2018), the approximate GP is then given as

$$f(x) \sim \mathcal{GP}\left(0, \frac{\sigma^2}{M}\phi(x)^T \phi(x')\right) = \mathcal{GP}\left(0, \tilde{k}(x - x')\right)$$

with an equivalent parametric expression

$$f(x) = \overline{\boldsymbol{a}}\phi(x),$$

where $\overline{\boldsymbol{a}}$ is a row vector as before, $p(\overline{\boldsymbol{a}}^T) = \mathcal{N}(\boldsymbol{0}, \boldsymbol{S})$ with $\boldsymbol{S} = \text{diag}(s(\omega_1), \ldots, s(\omega_M), s(\omega_1), \ldots, s(\omega_M))$ for regular Fourier features, and $p(\overline{\boldsymbol{a}}^T) = \mathcal{N}\left(\boldsymbol{0}, \frac{\sigma^2}{M}I\right)$ for random Fourier features.

Consider a GP prior VAE with a stationary covariance conditioned with a single continuous covariate $x$. Existing GP-VAE models assume the prior for the latent space factorizes across the dimensions, i.e., $p(\boldsymbol{z}|x) = \prod_{l=1}^{L} \mathcal{GP}(0, k_l(x, x'))$. For the reduced-rank approximation of a GP prior with $M$ Fourier features, the GP prior VAE can be directly implemented by LMM-VAE by setting $\boldsymbol{z} = A\boldsymbol{\phi}(x)$, where

$$A = \left(\bar{\boldsymbol{a}}_1^T, \ldots, \bar{\boldsymbol{a}}_L^T\right)^T,$$

$p(A) = \prod_{l=1}^{L} p(\bar{\boldsymbol{a}}_l^T)$ as defined above, and letting $\sigma_z^2 \to 0$. A similar connection between the LVAE (Ramchandran et al., 2021) and LMM-VAE can also be established (see Appendix F).

While connected to the GP-VAEs, LMM-VAE simplifies the overall training procedure. It achieves this via the spectral representation and global parameterization allowing for straightforward SGD optimization.

## 7    Experiments

We evaluate LMM-VAE on three tasks: (1) missing value imputation, (2) future prediction, and (3) time-based interpolation, measuring its performance using the Mean Squared Error (MSE). Our primary comparison is with the GP prior VAEs, which are most similar to our approach in modelling the prior through regression techniques. For consistency, we employed similar encoder and decoder architectures across all methods (see Appendix M for further details).

We rely on Health MNIST and Rotating MNIST, which are two datasets derived from MNIST (LeCun et al., 2010), and the Physionet Challenge 2012 dataset (Silva et al., 2012). See Appendices H, I and K for hyperparameters and further details on the experimental set-up not discussed in the main paper. Results in **bold** are within one standard deviation of the best mean per experimental set-up. As a consistency check, we also compared the performance of LMM and LMM-VAE on a simplified, low-dimensional set-up, which is presented in Appendix E.

### 7.1    Health MNIST

Following prior work (Ramchandran et al., 2021), we generate a longitudinal dataset with missing pixels by augmenting the MNIST dataset. This dataset replicates various characteristics present in real medical data, where each snapshot of the time series corresponds to the health state of a patient. The trajectory is designed to evolve non-linearly with time. We select the digits '3' and '6' to represent two biological sexes. There are $Q = 6$ covariates describing the dataset, which are age, id, diseasePresence, diseaseAge, sex, and location. Further details on the dataset and train-validation-test splits are in Appendix H.1.

By design, GPP-VAE and SVGP-VAE are parametrized by two kernels describing the object and view. As such, we train these two models using the id and age covariates per respective kernel. We parametrize LVAE using the optimal set-up as reported in their paper, i.e. id, age, sex×age, and diseasePresence×diseaseAge (Ramchandran et al., 2021). We train three LMM-VAE variants to demonstrate different parameterizations, summarizing the covariates for each configuration in Figure 3. Note that the color-coding in Table 2 follows this mapping.

**Missing Value Imputation.**    We report missing value imputation performance based on the *training* set in Table 2. LMM-VAE's imputation MSE remains robust across the different linear model parameterizations and outperforms the GP-based baselines.

**Future Prediction.**    We further evaluate our proposed model by its ability to generate new data, based on the *test* MSEs for conditional generation per model configuration. The predicted trajectories of two instances are illustrated in Appendix H.3.

By design, the data generation mechanism is a complex function of several key covariates. As shown in Figure 3, LMM-VAE's ability to model these covariates leads to a lower conditional test MSE compared to GPP-VAE and SVGP-VAE. In addition, LMM-VAE's sensitivity to different prior parametrizations highlights the importance of including relevant covariates that could reveal information about the data generation

Table 2: Imputation MSEs on Health MNIST.

| MODEL ($\dim(\boldsymbol{z}) = 32$) | MSE ↓ |
|---|---|
| GPP-VAE (CASALE ET AL., 2018) | $0.021_{\pm 0.0012}$ |
| SVGP-VAE (JAZBEC ET AL., 2021) | $0.015_{\pm 0.0012}$ |
| LVAE (RAMCHANDRAN ET AL., 2021) | $0.018_{\pm 0.0006}$ |
| oLMM-VAE, VI (OURS) | $\mathbf{0.002}_{\pm 0.0000}$ |
| oLMM-VAE, VI (OURS) | $\mathbf{0.002}_{\pm 0.0000}$ |
| oLMM-VAE, VI (OURS) | $\mathbf{0.002}_{\pm 0.0000}$ |

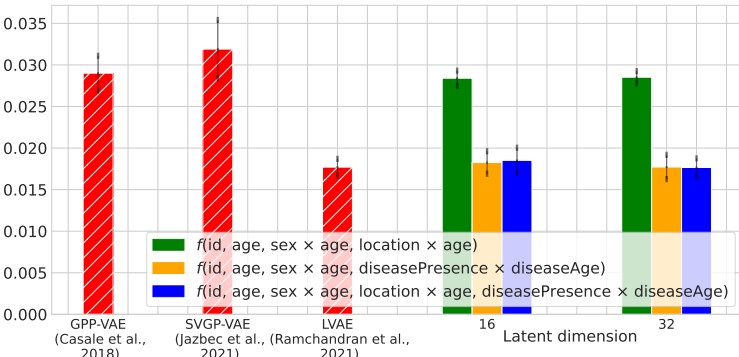

Figure 3: Predictive test MSEs on the Health MNIST dataset with latent dimension $\dim(\boldsymbol{z}) = 32$.

process. Compared to LVAE, Figure 3 indicates that LMM-VAE reports competitive MSE when the covariates used for training are identical.[2] Interestingly, the flexibility provided by a GP-prior may not translate into significant performance gain.

## 7.2 Rotating MNIST

The Rotating MNIST is another popular dataset used to benchmark CVAE models (Casale et al., 2018; Jazbec et al., 2021; Ramchandran et al., 2021). This dataset is composed of 400 unique instances of the digit '3' rotated through 16 angles uniformly distributed in $[0, 2\pi)$. Our treatment of the train-validation-test splits differs from that originally featured in (Casale et al., 2018; Jazbec et al., 2021), which we elaborate on in Appendix I.1.

**Interpolation.** In this section, we evaluate the models by their ability to *interpolate* four consecutive angles that are unseen for test subjects during training. The data generation mechanism depends only on the digit instance and angular information, which can be easily modelled by the GP priors. Consequently, GP-based VAEs are competitive baselines. Due to the deterministic nature of the data generation process, GSNN training yields superior performance. Still, sLMM-VAE is also able to match, and improve upon, the performance of its GP counterparts (see Table 3).

**Improving performance with basis functions.** In their basic form, LMMs may have limited modelling capacity. However, as discussed in Section 6, we can make LMM-VAE arbitrarily complex to suit the modelling task at hand. Using basis functions defined by regular Fourier frequencies (sin and cos with increasing frequencies $\omega_m$) for both LMM-VAE variants, we show in Figure 4 that test MSE for Rotating MNIST decreases or remains stable as the number of basis functions increases. This aligns with the theoretical considerations in Section 6, that adding basis functions to LMM-VAE makes it a strong, efficient approximation for GP prior VAEs, while preserving global parametrization and efficient training.

---

[2]LVAE contains an additional interaction term between id and age, see Ramchandran et al. (2021).

Table 3: Interpolation test MSEs on Rotating MNIST.

| MODEL (dim($z$) = 16) | MSE ↓ |
|---|---|
| CVAE (SOHN ET AL., 2015) | $0.036_{\pm 0.0033}$ |
| GPP-VAE (CASALE ET AL., 2018) | $0.017_{\pm 0.0005}$ |
| SVGP-VAE (JAZBEC ET AL., 2021) | $0.018_{\pm 0.0002}$ |
| LVAE (RAMCHANDRAN ET AL., 2021) | $0.026_{\pm 0.0007}$ |
| oLMM-VAE, VI (OURS) | $0.027_{\pm 0.0003}$ |
| sLMM-VAE, VI (OURS) | $0.017_{\pm 0.0003}$ |
| oLMM-VAE, GSNN (OURS) | $\mathbf{0.014}_{\pm 0.0003}$ |
| sLMM-VAE, GSNN (OURS) | $\mathbf{0.014}_{\pm 0.0004}$ |

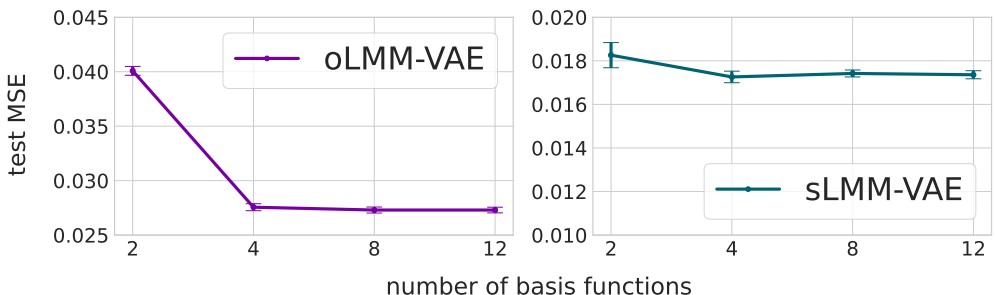

Figure 4: The test MSE decreases as the number of included basis functions increases (dim($z$) = 16).

## 7.3 Physionet Challenge 2012

**Future Prediction.** We also evaluate LMM-VAE on real-world medical data from the Physionet Challenge 2012 (Silva et al., 2012). The full dataset contains approximately 12000 patients monitored in the Intensive Care Unit for 48 hours. We modelled 37 different attributes, such as heart rate and body temperature. The auxiliary covariates used are *time of measurement, id, age, gender, icu type*, and *mortality*. Given the high dimensionality of the auxiliary covariates, we focus on LMM-VAE and LVAE, the two model classes designed to handle arbitrarily many covariates. As shown in Table 4, both LMM-VAE and LVAE effectively extract information from the auxiliary covariates for future prediction. The LMM-VAE variants trained with GSNN demonstrate competitive performance compared to LVAE, with further improvements observed as the number of basis functions increases in Figure 5. These results underscore their potential for real-world healthcare applications.

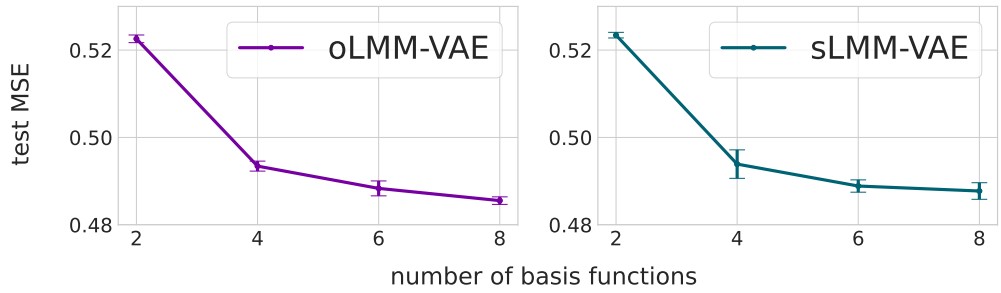

Figure 5: The test MSE decreases as the number of basis functions for the auxiliary covariate *time* increases. Plots are generated via GSNN training.

Table 4: Test MSEs on the Physionet Challenge 2012.

| MODEL ($\dim(\boldsymbol{z}) = 30$) | MSE $\downarrow$ |
|---|---|
| MEAN PREDICTION | $0.902_{\pm 0.0000}$ |
| VAE (KINGMA & WELLING, 2014) | $0.905_{\pm 0.0030}$ |
| LVAE (RAMCHANDRAN ET AL., 2021) | $\mathbf{0.480}_{\pm \mathbf{0.0349}}$ |
| oLMM-VAE, VI (OURS) | $0.618_{\pm 0.0052}$ |
| sLMM-VAE, VI (OURS) | $0.664_{\pm 0.0039}$ |
| oLMM-VAE, GSNN (OURS) | $\mathbf{0.486}_{\pm \mathbf{0.0009}}$ |
| sLMM-VAE, GSNN (OURS) | $\mathbf{0.488}_{\pm \mathbf{0.0019}}$ |

## 8  Conclusion and Future Work

We present LMM-VAE, a novel method for modelling high-dimensional longitudinal data that scales to large datasets with numerous covariates. Empirical evaluation reveals the importance of modelling auxiliary covariates and the potential of using LMM-VAE for longitudinal analyses. Theoretical analysis demonstrates connections to the GP-based VAEs. This connection provides a solid foundation to adapt LMM-VAE to different modeling tasks by adaptively incorporating basis functions, as well as establishes LMM-VAE's potential to be viewed as a reduced-rank approximation method for GP prior VAEs. Avenues for future research include extending our method to facilitate counterfactual reasoning, and incorporating alternative priors for the latent variables.

## Acknowledgements

We thank the Aalto Science-IT project for the generous computational resources. This work is supported by the Research Council of Finland (decision number: 359135).

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

# APPENDIX

## A    Connecting LMM Set-ups to LMM-VAE

To enhance clarity, we present a simple example that links the formulation of LMM set-ups to the context of LMM-VAE.

Consider a medical dataset where we aim to model the latent space using covariates such as patient ID, age, and time. For simplicity, assume the dataset includes three unique patients, and we apply one-hot encoding (OHE) to represent patient ID.

In the LMM-VAE framework, the response variable is the latent variable, $\mathbf{z} \in \mathbb{R}^L$. We categorize the covariates as follows: the effect of patient ID is modelled as a random effect, while that for age and time are treated as fixed effects. The $L$-dimensional latent variable for patient $p$ at time point $t$ can then be modelled as:

$$
\boldsymbol{z}_t^{(p)} = A_S \begin{pmatrix} \text{time} \\ \text{age} \end{pmatrix} + A_R \begin{pmatrix} 1 \\ 0 \\ 0 \end{pmatrix} + \boldsymbol{\epsilon}
$$

$$
= \left( A_S, A_R \right) \begin{pmatrix} \text{time} \\ \text{age} \\ 1 \\ 0 \\ 0 \end{pmatrix} + \boldsymbol{\epsilon}
$$

$$
= A\boldsymbol{x}_t^{(p)} + \boldsymbol{\epsilon}.
$$

Here, the fixed and random effects matrices, $A_S$ and $A_R$, respectively. $\boldsymbol{\epsilon}$ represents the noise or error term, and $\boldsymbol{x}_t^{(p)}$ is the vector of covariates (such as time, age, and one-hot encoded patient ID) of patient $p$ at timepoint $t$.

We can stack the observations for all time points of patient $p$ as:

$$
Z_p = AX_p + E_p,
$$

and more generally, the entire dataset can be represented as

$$
Z = AX + E,
$$

which we arrive at following our formalism in the paper. This approach provides a clear structure for modelling the latent space while accounting for both fixed and random effects through a global parametrization, using covariates that are relevant to the dataset.

## B    Hierarchical Plate Model

## C    sLMM-VAE

### C.1    sLMM-VAE ELBO formulation

Assuming a factorizing variational posterior $q(A, \theta, E_K | Y, X) = \prod_{k=1}^{K} q_\psi(\boldsymbol{\epsilon}^{(k)} | Y, A, X) q(A) q(\theta)$ where all distributions are Gaussians, we obtain a tractable ELBO similarly to that of oLMM-VAE in the main text

$$
\log p(Y|X) \geq \sum_{k=1}^{K} \sum_{n \in I_k} \Big( \mathbb{E}_{q(\theta) q_\psi(\boldsymbol{\epsilon}^{(k)}|Y,A,X)} \left[ \log p(\boldsymbol{y}_n | \theta, \boldsymbol{z}_n) \right]
$$

$$
- \mathbb{E}_{q(A)} \left[ \text{KL} \left( q_\psi(\boldsymbol{\epsilon}^{(k)} | Y, A, X) \, \| \, p(\boldsymbol{\epsilon}^{(k)}) \right) \right] \Big)
$$

$$
- \text{KL} \left( q(A) \, \| \, p(A) \right) - \text{KL} \left( q(\theta) \, \| \, p(\theta) \right),
$$

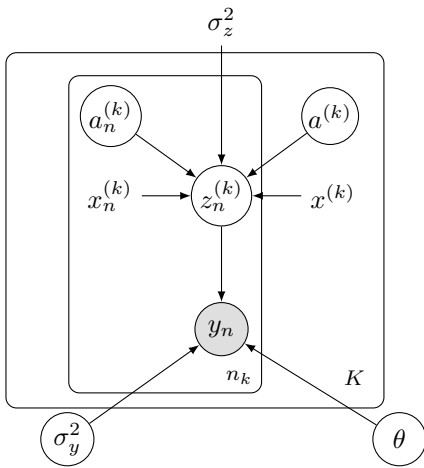

Figure 6: LMM-VAE's hierarchical plate diagram, which illustrates both fixed and random effects.

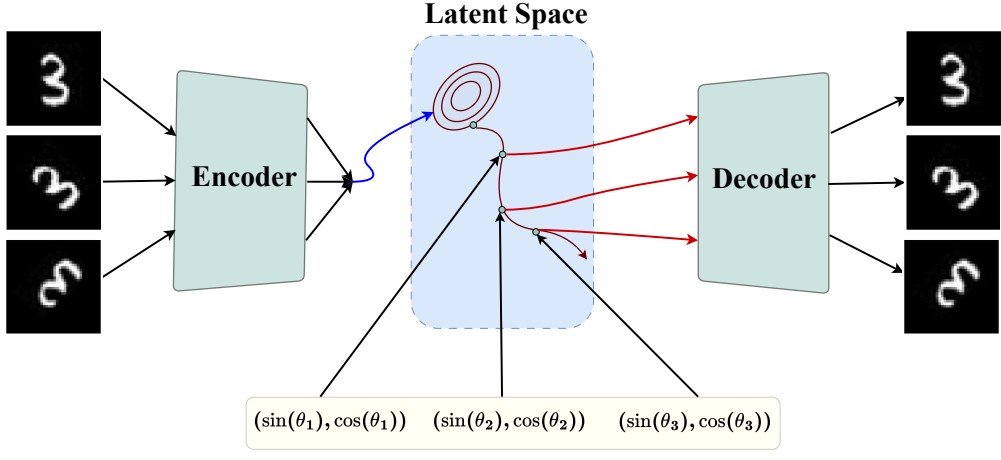

Figure 7: Diagram of sLMM-VAE.

where $\boldsymbol{z}_n = A\boldsymbol{x}_n + \boldsymbol{\epsilon}^{(k)}$. As with oLMM-VAE, we could drop prior distributions $p(\theta)$ and $p(A)$ and treat the generative model parameters as hyperparameters. We refer to this model class as the *structured LMM-VAE (sLMM-VAE)*. The associated diagram for sLMM-VAE can be found in Figure 7.

The formulation of sLMM-VAE could yield competitive performance when the auxiliary covariates sufficiently characterize the output. This intuition is outlined below:

1. The noise variable $\boldsymbol{\epsilon}$ is identical for observations arising from the same group, thus reducing stochasticity.

2. The linear model $A$ is now explicitly accounted for and learned as part of the decoder. Note that in the original oLMM-VAE, $A$ is only learned as part of the prior.

## D  Gaussian stochastic neural network training objective for LMM-VAEs

We credit Sohn et al. (2015) for introducing the Gaussian stochastic neural network (GSNN) training objective in the context of conditional generation. We expand on the motivation behind this approach. First, the ELBO objective consists of the well-known reconstruction and regularization (the KL divergence) terms. This means that at training time the data $\boldsymbol{y}$ is reconstructed from the variational approximation, whereas at test time, the data is generated from the prior. The key insight from Sohn et al. (2015) was to ensure consistency

between training-time reconstruction and test-time generation. To achieve this, the authors propose aligning the variational approximation with the prior, i.e., setting $q_\phi(\boldsymbol{z}_n|\boldsymbol{y}_n) = p(\boldsymbol{z}_n|A, \boldsymbol{x}_n)$. Substituting this into the ELBO results in the KL divergence becoming exactly zero, reducing the ELBO to the GSNN training objective as shown in in Eq. 11 in Section 4.4. In essence, the GSNN training objective corresponds to the standard ELBO when the variational approximation equals the prior.

The GSNN training framework is suitable for conditional generation when the covariates $\boldsymbol{x}_n$ are rich enough such that they can sufficiently guide the generation of the corresponding data sample $\boldsymbol{y}_n$. In other words, when the covariates $\boldsymbol{x}_n$ are sufficiently informative in characterizing $\boldsymbol{y}_n$, the GSNN training objective explicitly trains the model to generate accurate/high-quality data samples $\boldsymbol{y}_n$, leading to competitive performance. Results from our empirical evaluation supports this statement.

GSNN can also be understood from the regression view point: if the amount of variability in the latent variables approaches zero, then GSNN training objective would effectively reduce to regression. To gain further intuition about what the training procedure does for the LMM-VAE model, we can explicitly re-write the GSNN loss as

$$
\begin{aligned}
\mathcal{L}_{\text{GSNN}}(Y|X, A, \theta) &= \frac{1}{S} \sum_{s=1}^{S} \sum_{n=1}^{N} \log p(\boldsymbol{y}_n|\boldsymbol{z}_{n,s}, \theta) \\
&= \frac{1}{S} \sum_{s=1}^{S} \sum_{n=1}^{N} \log p(\boldsymbol{y}_n|A\boldsymbol{x}_n + \boldsymbol{\epsilon}_{n,s}, \theta).
\end{aligned}
$$

We now observe that the trained network is composed of a first linear layer conditioned on $\boldsymbol{x}_n$, where there is some noise injection (can also be thought of as regularization), followed by another network parameterizing the decoder. In other words, we are now learning a mapping from $\boldsymbol{x}_n \to \boldsymbol{y}_n$ via a neural network composed of a linear layer and a second network. When we rewrite the loss in the aforementioned manner, there is a strong resemblance to the task of regression (with some regularization in the first layer). That is, we are explicitly training the linear model together with the decoder network to maximize the expected log-likelihood of conditioned on . This stands in contrast to variational inference, where we learn indirectly by primarily minimizing $\mathbb{E}_{q(A)}\big[KL\big(q_\psi(\boldsymbol{z}_n|\boldsymbol{y}_n)||p(\boldsymbol{z}_n|A, \boldsymbol{x}_n)\big)\big]$.

Additionally, we would like to point out that a limitation of the GSNN objective is that it does not provide any estimate of the latent variables because it is designed for the generative task only.

## E  Further characterization of LMM-VAE

### E.1  Additional comparison with SVGP-VAE

The approximation quality of LMM-VAE improves as the number of basis functions increases. Similarly, the sparsity of SVGP-VAE is determined by the number of inducing points, with a larger set generally leading to a richer approximation. In an ablation study, we analyze the effect of varying the number of inducing points in SVGP-VAE using the rotating MNIST experiments, and include the results in Table 5. We observe an improvement in Test MSE as the number of inducing points increases, leading to a consistent, albeit gradual, enhancement.

A similar trend is observed with sLMM-VAE Figure 4, where test performance improves and stabilizes as more basis functions are added. These results indicate alignment in trends across the different GP approximation methods. While SVGP-VAE and sLMM-VAE deliver comparable and competitive results, the GSNN variants of LMM-VAE outperform both, achieving the strongest test performance in the rotating MNIST experiments.

### E.2  LMMs vs LMM-VAEs

To better characterize LMM-VAE's performance, we evaluate its behaviour in two low-dimensional settings: one where classical linear mixed models (LMMs) are expected to perform well, and another where their performance is likely to be limited.

Table 5: Test MSE as a function of inducing points for SVGP-VAE on Rotating MNIST.

| MODEL ($\dim(\boldsymbol{z}) = 16$) | NO. OF INDUCING POINTS | MSE ↓ |
|---|---|---|
| SVGP-VAE (JAZBEC ET AL., 2021) | 16 | $0.019_{\pm 0.0005}$ |
| | 32 | $0.018_{\pm 0.0002}$ |
| | 48 | $0.017_{\pm 0.0003}$ |
| | 64 | $0.017_{\pm 0.0007}$ |

We simulate a low-dimensional synthetic dataset based on a latent variable model. A univariate latent variable is defined for each subject across time, incorporating a subject-specific random intercept and a fixed effect for time. This latent variable drives three observed variables $\boldsymbol{y} \in \mathbb{R}^3$ via predefined linear measurement functions, with additive Gaussian noise. The resulting dataset captures structured variation across subjects and over time, providing a controlled environment for model comparison. The experiments were conducted as a function of the number of subjects, each observed at 20 timepoints. For validation and testing, the last 5 timepoints from 20 subjects were held out, with the remaining data used for training.

We compare two modelling approaches: a classical linear mixed model (LMM) and a variant of oLMM-VAE, where the observed response is modelled as a linear transformation of the latent variable. Based on the test MSE in the first column of Figure 8, both models exhibit similar trends with respect to dataset size (i.e., number of subjects). This outcome aligns with expectations, as both models are well-suited to linear generative processes. However, the oLMM-VAE achieves lower overall test MSE, suggesting improved generalization.

To further evaluate the oLMM-VAE's flexibility, we generate a second dataset variant in which $\boldsymbol{y}$ is a now non-linear functions of the latent variable. We train both the LMM and oLMM-VAE to this new dataset and report the results in the second column of Figure 8. In this set-up, the oLMM-VAE is extended to incorporate a non-linear decoder, enabling it to learn a flexible mapping from the latent space to the observed outputs. With this added capacity, the oLMM-VAE yields substantially lower test MSE than the classical LMM, highlighting its advantage in capturing non-linear structure.

### E.2.1 Experimental details

LMM was implemented via the lme4 library in R (Bates et al., 2015), following the approach described in `https://mac-theobio.github.io/QMEE/lectures/MultivariateMixed.notes.html#a-trick-to-do-multivariate-mixed-models-using-lme4` to fit multivariate mixed models.

Meanwhile, the oLMM-VAE was employed and trained using GSNN training. For the linearly generated dataset, the decoder was parameterized as a linear function and trained with a learning rate of 0.001. For the non-linearly generated data, the decoder consisted of two hidden layers with Tanh nonlinearities and hidden sizes of 16 and 8, trained with a learning rate of 0.01. Across both set-ups, we use a latent dimension of 1, set $\sigma_z$ as 0.005, and use the AdamW Optimizer (Loshchilov & Hutter, 2019). We monitored the validation loss and applied a strategy akin to early stopping, saving the model weights with the lowest validation loss. The oLMM-VAE was trained for a maximum of 2000 epochs.

## F GP prior VAEs as LMM-VAEs

Here, we outline how LVAE can be implemented as LMM-VAE. Assuming the $j$-th additive latent component in LVAE model depends on a specific subset of covariates $\boldsymbol{x}^{(j)} \in \mathcal{X}^{(j)} \subset \mathcal{X}$ and that the corresponding Fourier features are $\boldsymbol{\phi_j}(\boldsymbol{x}^{(j)})$. The linear basis function approximation for the $j$-th additive latent component is then $\boldsymbol{z}^{(j)} = A^{(j)}\boldsymbol{\phi}_j(\boldsymbol{x}^{(j)})$ and the complete model can be written as

$$\boldsymbol{z} = \sum_{j=1}^{R} \boldsymbol{z}^{(j)} = \sum_{j=1}^{R} A^{(j)}\boldsymbol{\phi}_j(\boldsymbol{x}^{(j)}), \tag{15}$$

where $R$ is the number of additive kernels in an LVAE model and $p(A) = \prod_{j=1}^{R} p(A^{(j)})$.

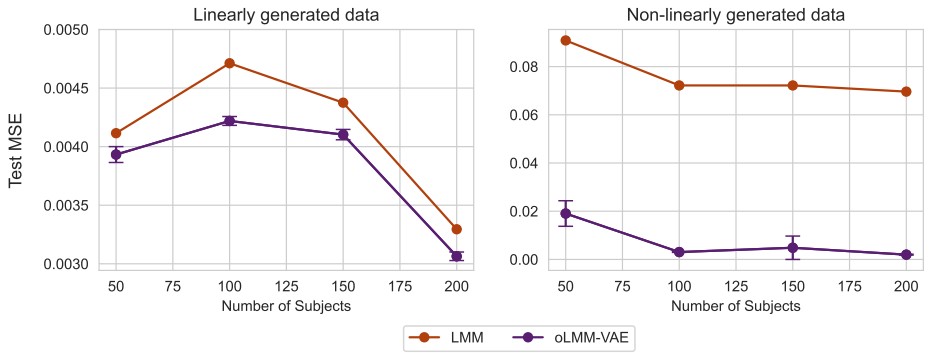

Figure 8: Test MSEs, averaged across dimensions, as a function of the number of subjects, based on simulated data. The first column describes the set-up in which $\boldsymbol{y}$ is generated as a linear function of the latent variable, while the second column describes that in which $\boldsymbol{y}$ is non-linear function of the latent variable.

## G    Identifiability

Given that our model belongs to the family of conditional VAEs, LMM-VAE inherits the property of being identifiable. This result was established by Khemakhem et al. (2020), who rely on auxiliary covariates $\boldsymbol{x}$ to constrain the latent space via a conditional prior $p(\boldsymbol{z}|\boldsymbol{x})$.

**Formulation.**    In general, an unconstrained latent-variable model of the form

$$p(\boldsymbol{y}, \boldsymbol{z}) = p(\boldsymbol{y}|\boldsymbol{z})p(\boldsymbol{z}),$$

is not identifiable. Instead Khemakhem et al. (2020) assume, adapted to our notation, that

$$p(\boldsymbol{y}, \boldsymbol{z}|\boldsymbol{x}) = p_f(\boldsymbol{y}|\boldsymbol{z})p_{T,\lambda}(\boldsymbol{z}|\boldsymbol{x}),$$

where $p_f(\boldsymbol{y}|\boldsymbol{z}) = f(\boldsymbol{z}) + \varepsilon$, with $\varepsilon \sim p(\varepsilon)$, and $f : \mathcal{Z} \to \mathcal{Y}$ is an injective mapping, e.g., a neural net. The conditional prior

$$p_{T,\lambda}(\boldsymbol{z}|\boldsymbol{x}) = \prod_{i=1}^{\dim(\boldsymbol{z})} m_i(z_i) \exp\Big[ \sum_{j=1}^{k} T_{ij}(z_i)\lambda_{ij}(\boldsymbol{x}) \Big],$$

is a fully factorized member of the exponential family of distributions. Here, $T_{ij}$ are the $k$ sufficient statistics, and $\lambda(\cdot)$ some function of $\boldsymbol{x}$, e.g., a neural net.

Relevant to our current discussion are Definitions 1 and 2 of Khemakhem et al. (2020), which define our notion of identifiability. We cite here in full for completeness.

**Definition 1 (Khemakhem et al., 2020).**    *Let $\sim$ be an equivalence relation on $\Theta$. We say that $p(\boldsymbol{x}|\boldsymbol{z})p(\boldsymbol{z})$ is identifiable up to $\sim$ if*

$$p_\theta(\boldsymbol{x}) = p_{\tilde{\theta}}(\boldsymbol{x}) \Rightarrow \theta \sim \tilde{\theta}.$$

*The elements of the quotient space $\Theta/_\sim$ are called the identifiability classes.*

**Definition 2 (Khemakhem et al., 2020).**    *Let $\sim$ be the equivalence relation on $\Theta$ defined as follows:*

$$(f, \boldsymbol{T}, \lambda) \sim (\tilde{f}, \tilde{\boldsymbol{T}}, \tilde{\lambda}) \Leftrightarrow \exists A, \boldsymbol{c} \quad (f^{-1}(x)) = A\tilde{\boldsymbol{T}}(\tilde{f}^{-1}(\boldsymbol{x})) + \boldsymbol{c} \quad \forall \boldsymbol{x} \in \mathcal{X},$$

*where $A$ is a $\dim(\boldsymbol{z})k \times \dim(z)k$ matrix and $\boldsymbol{c}$ is a vector. If $A$ is* invertible*, we denote this relation by $\sim_A$. If $A$ is a* block permutation *matrix, we denote it by $\sim_P$.*

Given these, the main theorem is

**Theorem 1 (Khemakhem et al., 2020).** *Assume that we observe data sampled from a generative model defined according to the model specified above with parameters $(f, \boldsymbol{T}, \lambda)$. Assume the following holds:*

*(i) The set $\{\boldsymbol{x} \in \mathcal{X} | \varphi_\varepsilon(\boldsymbol{x}) = 0\}$ has measure zero, where $\varphi_\varepsilon$ is the characteristic function of $p(\varepsilon)$.*

*(ii) the mixing function $f$ is injective.*

*(iii) the sufficient statistics $T_{ij}$ are differentiable almost everywhere and $(T_{i,j})_{1 \le j \le k}$ are linearly independent on any subset of $\mathcal{X}$ of measure greater than zero.*

*(iv) there exist $\dim(\boldsymbol{z})k + 1$ distinct points $\boldsymbol{x}^0, \dots, \boldsymbol{x}^{\dim(\boldsymbol{z})k}$ such that the matrix*

$$L = \left(\lambda(\boldsymbol{x}^1) - \lambda(\boldsymbol{x}^0), \dots, \lambda(\boldsymbol{x}^{\dim(\boldsymbol{z})k}) - \lambda(\boldsymbol{x}^0)\right),$$

*of size $\dim(\boldsymbol{z})k \times \dim(\boldsymbol{z})k$ is invertible,*

*then the parameters $(f, \boldsymbol{T}, \lambda)$ are $\sim_A$-identifiable.*

Picking $p(\varepsilon) = \mathcal{N}(\varepsilon | 0, \sigma_\varepsilon^2)$ and a suitable choice of architecture for $f$ all that remains to be shown is that our prior is a mean-field exponential family distribution. We have that for a normal prior $p(A) = \mathcal{N}(\text{vec}(A) | 0, \beta^{-1})$

$$
\begin{aligned}
p(\boldsymbol{z}|\boldsymbol{x}) &= \int p(\boldsymbol{z}|A, \boldsymbol{x})p(A)dA \\
&= \int \mathcal{N}(\boldsymbol{z}|A\boldsymbol{x}, \sigma_z^2 I)p(\text{vec}(A)|0, \beta^{-1})dA \\
&= \prod_{i=1}^{\dim(\boldsymbol{z})} \mathbb{E}_{\mathcal{N}(\bar{\boldsymbol{a}}_i|0, \beta^{-1})} \left[ \mathcal{N}\left(z_i | \bar{\boldsymbol{a}}_i \boldsymbol{x}, \sigma_z^2\right) \right] \\
&= \prod_{i=1}^{\dim(\boldsymbol{z})} \mathcal{N}\left(z_i | 0, \sigma_z^2 + \boldsymbol{x}^T \boldsymbol{x} \beta^{-1}\right),
\end{aligned}
$$

i.e., $p(\boldsymbol{z}|\boldsymbol{x})$ is an exponential family distribution. Assuming a rich enough variational posterior $q(\boldsymbol{z})$ we can then guarantee asymptotic identifiability given Theorem 4 by Khemakhem et al. (2020).

## G.1 Clinical Trial Data and Health MNIST

For brief insight into LMM-VAE's identifiability, we perform a series of experiments involving a randomized control trial (RCT) dataset based on colorectal cancer treatment Green et al. (2015), and Health MNIST. Additional information about the RCT data is disclosed in Appendix K.

**Illustrating Identifiability.** A measure to evaluate identifiability is the mean correlation coefficient (MCC) between the true latent variables and their estimated posterior (Hyvärinen & Morioka, 2016; Khemakhem et al., 2020). Computing the MCC[3] necessitates access to the ground truth for correlation comparisons, an unrealistic assumption in practice. A common approximation method involves comparing the correlation between different training runs, together with a second metric evaluating the latent space's predictive performance. While desirable, obtaining identifiability is still conditioned on good predictive performance, as a latent space that collapses to the same bad local optimum per run would still result in perfect correlation.

We compute one variant of LMM-VAE's MCC alongside a predictive metric (NLL or MSE). We include at least one other model per experimental set-up to serve as a baseline. For both real world clinical trial data and Health MNIST, LMM-VAE reports competitive results across multiple latent dimensions (see Tables 6 and 7, respectively).

---

[3]We use Khemakhem et al. (2020)'s implementation to compute the MCC: `https://github.com/siamakz/iVAE/blob/master/lib/metrics.py`.

Table 6: MCC for oLMM-VAE and VAE on the clinical trial data.

| Model | Latent Dimension | NLL ↓ | MCC ↑ |
|---|---|---|---|
| VAE | 20 | $-10.0_{\pm 8.3}$ | $0.47_{\pm 0.03}$ |
| oLMM-VAE (ours) | | $-14.1_{\pm 2.9}$ | $0.50_{\pm 0.03}$ |
| VAE | 18 | $-10.9_{\pm 7.5}$ | $0.48_{\pm 0.05}$ |
| oLMM-VAE (ours) | | $-10.7_{\pm 6.7}$ | $0.51_{\pm 0.02}$ |
| VAE | 15 | $-9.91_{\pm 4.8}$ | $0.51_{\pm 0.03}$ |
| oLMM-VAE (ours) | | $-11.6_{\pm 1.4}$ | $0.53_{\pm 0.03}$ |

Table 7: MCC for LMM-VAE and VAE on Health MNIST. We use the orange parametrization for oLMM-VAE.

| Model | Latent Dimension | MSE ↓ | MCC ↑ |
|---|---|---|---|
| VAE | 32 | $0.059_{\pm 0.001}$ | $0.59_{\pm 0.03}$ |
| oLMM-VAE (ours) | | $0.023_{\pm 0.000}$ | $0.58_{\pm 0.01}$ |
| VAE | 16 | $0.058_{\pm 0.002}$ | $0.57_{\pm 0.02}$ |
| oLMM-VAE (ours) | | $0.023_{\pm 0.001}$ | $0.56_{\pm 0.02}$ |
| VAE | 8 | $0.056_{\pm 0.001}$ | $0.62_{\pm 0.04}$ |
| oLMM-VAE (ours) | | $0.025_{\pm 0.001}$ | $0.60_{\pm 0.03}$ |

## H   Health MNIST

### H.1   Dataset Description

We rely on the dataset construction script from Ramchandran et al. (2021).

To generate a shared age-related effect, we gradually shift all digit instances towards the right corner over time. Half of the instances of '3' and '6' are assumed to be healthy (diseasePresence = 0), while the other half are inflicted with a disease (diseasePresence = 1). The diseased instances are rotated across 20 timepoints, with the rotation degree determined by the time to disease diagnosis (diseaseAge).

We further include a binary noise covariate location, which is randomly assigned to each unique instance, and apply a random rotational jitter to each data point to simulate noisy observations. Additionally, we mask out 25% of each image's pixels (to assess imputation capabilities).

In total, there are 1300 unique instances present in the dataset, where 650 correspond to the biological sex Male, and the remaining 650 correspond to Female. Each of these unique instances have a sequence length of 20. We withhold the last 15 timepoints of 100 subjects to construct the test set. The first five timepoints of these aforementioned subjects are included in the training set. The remaining dataset is then randomly split to construct the train and validation sets, in an approximate ratio of 85 : 15.

### H.2   Experimental details

We took the implementation of the baseline SVGP-VAE Jazbec et al. (2021) from `https://github.com/ratschlab/SVGP-VAE`, GPP-VAE Casale et al. (2018) from `https://github.com/fpcasale/GPPVAE`, and LVAE Ramchandran et al. (2021) from `https://github.com/SidRama/Longitudinal-VAE`. To perform experiments, we use the default hyperparameter setting specified in the respective repositories. Across all baselines, the architecture used for the experiment can be found in Appendix M.

For experiments regarding LMM-VAE, we use Adam optimizer Kingma & Ba (2015) learning rate of 0.001. We monitor the loss on the validation set and employ a strategy similar in spirit to early stopping, where we

save the weights of the model with the optimal validation loss. LMM-VAE was allowed to run for a maximum of 2500 epochs. We define $\sigma_z = 1$.

For all experiments, we report the mean and standard deviation obtained across five runs.

### H.3 Additional illustrations of LMM-VAE's predictions

We visualize two sets of trajectories corresponding to 2 individuals in Figure 9.

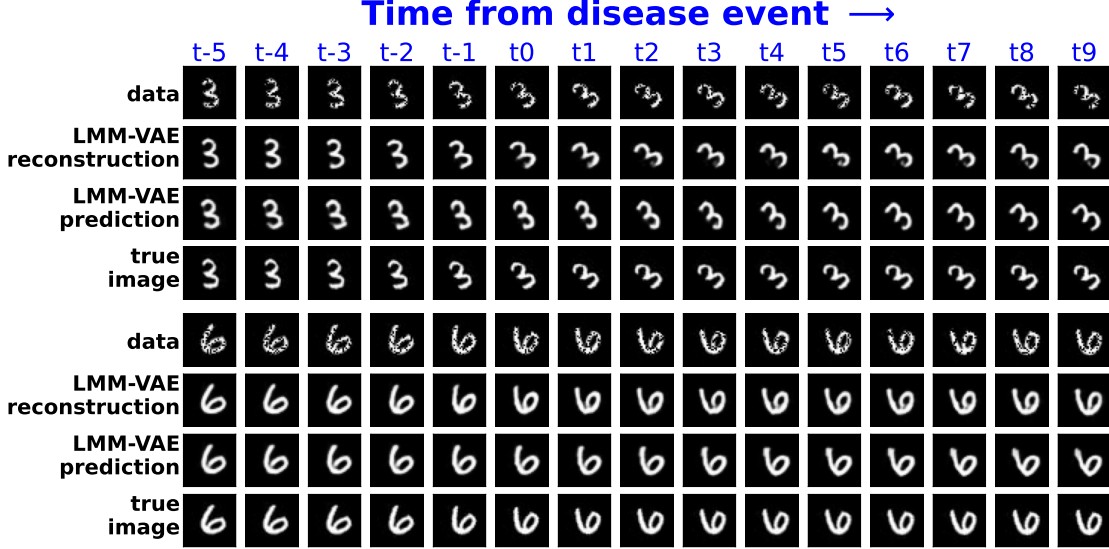

Figure 9: We illustrate 2 sets of images here, each corresponding to a different biological sex. Per image set, we visualize the reconstructions and predictions obtained on the test set by oLMM-VAE with 16 latent dimensions in the second and third rows. The noisy data along with the original uncorrupted images are also depicted in the first and last rows, respectively.

### H.4 Supplementary tables for Health MNIST

The tables containing the experimental results for Health MNIST are described in Table 8 and Table 9.

Table 8: Imputation MSEs for LMM-VAE on the HealthMNIST Dataset.

| Model | Latent Dimension | Imputation MSE ↓ |
|---|---|---|
| LMM-VAE | | $0.002_{\pm 0.0000}$ |
| LMM-VAE | 16 | $0.002_{\pm 0.0000}$ |
| LMM-VAE | | $0.002_{\pm 0.0000}$ |
| LMM-VAE | | $0.002_{\pm 0.0000}$ |
| LMM-VAE | 32 | $0.002_{\pm 0.0000}$ |
| LMM-VAE | | $0.002_{\pm 0.0000}$ |

## I Rotating MNIST

### I.1 Dataset Description

We revise the train/validation/test splits to reflect our interest in evaluating the learned id-specific parameters. This method of evaluation is not new Gelman (2006); Rabinowicz & Rosset (2022). Our modification results

Table 9: Predictive Test MSEs for LMM-VAE on the HealthMNIST Dataset.

| MODEL | LATENT DIMENSION | PREDICTIVE MSE ↓ |
|-------|------------------|------------------|
| LMM-VAE | | $0.0284_{\pm 0.0007}$ |
| LMM-VAE | 16 | $0.0183_{\pm 0.0010}$ |
| LMM-VAE | | $0.0185_{\pm 0.0014}$ |
| LMM-VAE | | $0.0285_{\pm 0.0009}$ |
| LMM-VAE | 32 | $0.0177_{\pm 0.0013}$ |
| LMM-VAE | | $0.0177_{\pm 0.0008}$ |

in the validation set comprising of ids that are also present in the train and test sets, which is not the case in the original set-up (Casale et al., 2018; Jazbec et al., 2021).

We generate 400 instances of the digit '3' and rotate each of these unique instances through 16 angles spaced uniformly in $[0, 2\pi)$. This results in 6400 images in the entire dataset. To construct the test set, we consider 80 instances at random, and take subsequences corresponding to four consecutive angles of the aforementioned digits. The train and validation sets are then randomly constructed based on the remaining images, in an approximate ratio of $80 : 20$.

## I.2 Experimental details

For the baselines CVAE (Sohn et al., 2015), GPP-VAE (Casale et al., 2018), and SVGP-VAE (Jazbec et al., 2021) we rely on the implementation from `https://github.com/ratschlab/SVGP-VAE`. The default experimental set-up is used, with a slight modification to the final activation layer (for purposes of standardizing the architectures across models). See Appendix M for the exact architecture used.

To perform experiments for LVAE, we use the experimental set-up from `https://github.com/SidRama/Longitudinal-VAE` (Ramchandran et al., 2021). Similar to the other baselines, the actual architecture used for the experiment can be found in Appendix M.

For experiments regarding LMM-VAE, we use the Adam optimizer (Kingma & Ba, 2015) and a learning rate of 0.001. We also use an exponential learning rate scheduler with a step size of 500. We monitor the loss on the validation set and employ a strategy similar in spirit to early stopping, where we save the weights of the model with the optimal validation loss. LMM-VAE was allowed to run for a maximum of 2000 epochs. We also define $\sigma_z = 1$.

For all experiments, we report the mean and standard deviation obtained across four runs.

## I.3 Additional illustrations on Interpretability

**Illustrating Interpretability.** As an additive model, LMM-VAE enjoys the benefit of being interpretable, which has merited interest in domains such as healthcare Plate (2000). We showcase this property with the sLMM-VAE variant on Rotating MNIST in Figure 10. Conditioned on an instance-specific $\boldsymbol{\epsilon}^{(k)}$, the latent space mapping is fully deterministic as a function of the auxiliary covariates.

Representing the rotation angle by $\theta$, the first two basis functions for the Rotating MNIST are $\boldsymbol{x} = (\sin(\theta), \cos(\theta))^T$. For latent dimension $l$, the latent space $z_l^{(k)} = \overline{\boldsymbol{a}}_l \boldsymbol{x} + \epsilon_l^{(k)}$, where $k$ denotes the digit instance under consideration. We illustrate both shared and random effects in Figure 10. The vertical shifting of the graph from the mean function, as induced by $\epsilon_l^{(k)}$, corresponds to id-level variation. Simultaneously, the rate of angular rotation remains consistent for the different digit ids, constituting a shared effect. Additional illustrations of how different noise realizations correspond to different digit identities can be found in Appendix I.3.

Additional illustrations (Figure 11 , Figure 12, Figure 13) further illustrate how different noise realizations correspond to different digit identities.

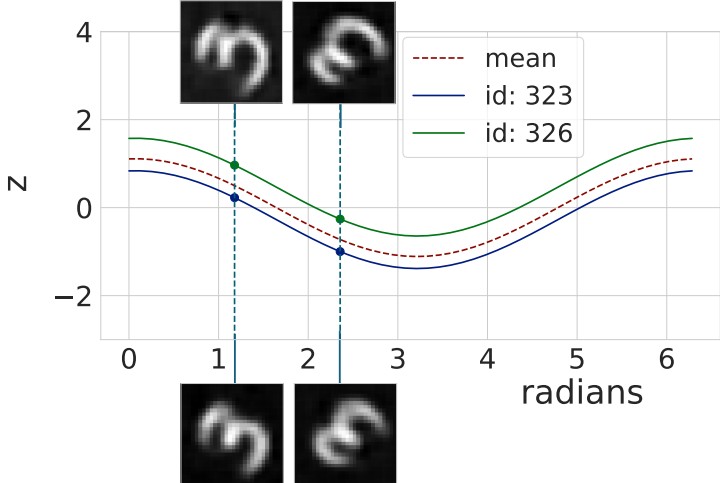

Figure 10: The dashed red line represents the expected function of the $1^{st}$ latent dimension w.r.t angular information. We obtain the trajectories of two specific instances (id: 323 and 326) by adding $\epsilon_1^{(323)}$ and $\epsilon_1^{(326)}$, respectively. Note that by moving along one of these deterministic trajectories, we get images of the same instance rotated through different angles.

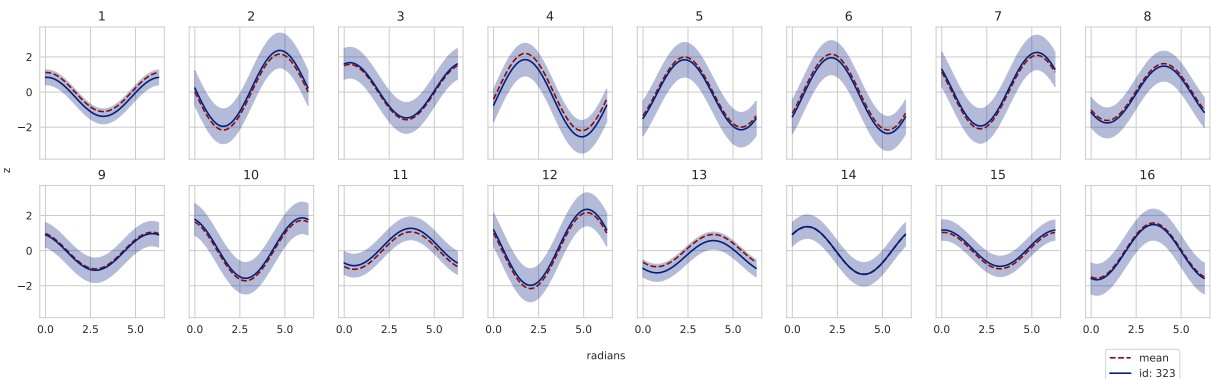

Figure 11: Visualisation of each of the 16 latent dimensions with their corresponding mean plot (represented by the dashed red line). In addition, the mean of the digit instance (id: 323) is also indicated here (solid blue line). The shaded region corresponds to the uncertainty in the function draw of id:323 (1 standard deviation is represented here).

### I.4 Further experiments for Rotating MNIST

Here, we provide additional illustrations (Figure 14, Figure 15) highlighting how additional basis functions can lead to improvement in performance.

## J Physionet Challenge 2012

### J.1 Dataset Description

In total, there are 3599 unique instances present in the dataset for training. This patient subset is selected such that each instance is represented by at least 15 observations. We withhold the last 10 timepoints of 1200 patients to construct the test set. The train and validation sets are then randomly constructed based on the remaining observations, in an approximate ratio of $80:20$. The resulting train and validation sets comprise of 3599 and 3598 unique instances respectively.

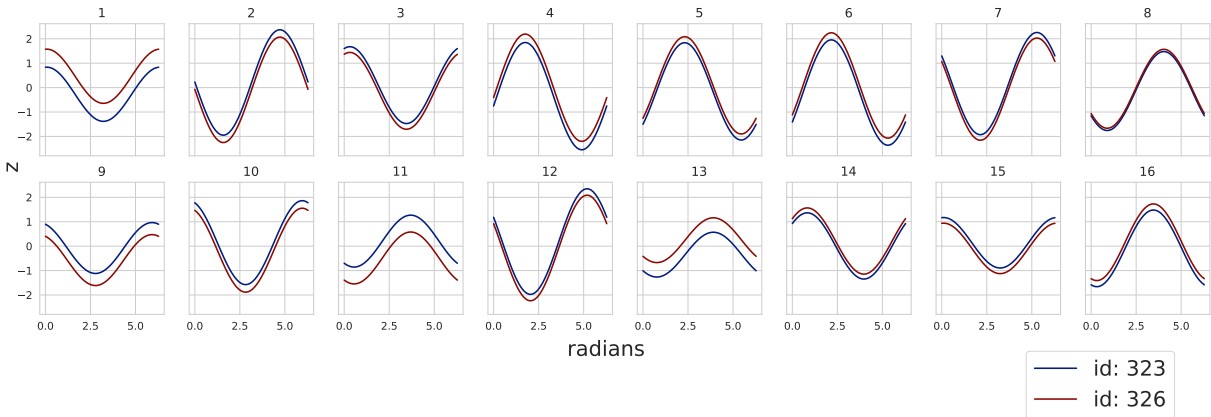

Figure 12: Visualisation of each of the 16 latent dimensions with the addition of the mean of id-specific noise. In this plot, 2 unique ids are represented.

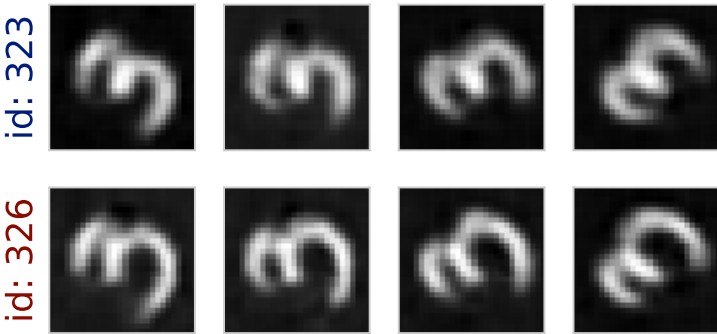

Figure 13: Test predictions of 2 different digit instances corresponding to Figure 12. The generated digits are rotated through the same angles. However, they assume different identities with different noise realizations along each latent dimension.

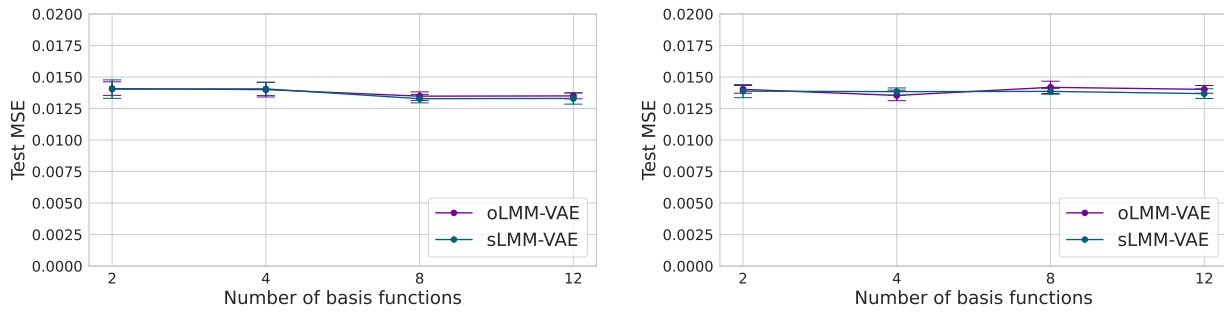

Figure 14: Plot generated via GSNN training of oLMM-VAE and sLMM-VAE. Left: Latent dimension of 32. Right: Latent dimension of 16. In general, MSE generally decreases as the number of included basis functions increases for latent dimension of 32. We assume $\theta$ and $A$ are deterministic in this set of experiments.

## J.2   Experimental details

We use the Adam optimizer (Kingma & Ba, 2015) and a learning rate of 0.001, with an exponential learning rate scheduler. We monitor the loss on the validation set and employ a strategy similar in spirit to early

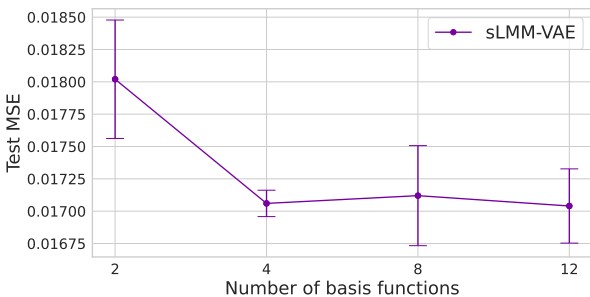

Figure 15: Plot generated via VI of sLMM-VAE for a latent dimension of 32. In general, MSE generally decreases as the number of included basis functions increases. We assume $\theta$ and $A$ are deterministic in this set of experiments.

stopping, where we save the weights of the model with an optimal validation loss. LMM-VAE was allowed to run for a maximum of 1000 epochs.

To perform experiments for LVAE, we use the experimental set-up from `https://github.com/SidRama/Longitudinal-VAE` (Ramchandran et al., 2021). Similar to the other baselines, the actual architecture used for the experiment can be found in Appendix M.

For all experiments, we report the mean and standard deviation obtained across five runs.

# K   Project Data Sphere

## K.1   Dataset Description

We use a medical time series dataset derived from a randomized control trial for the treatment of colorectal cancer (dataset identifier: Colorec SanfiU 2007 131). This dataset is taken from the open data sharing platform 'Project Data Sphere' (Green et al., 2015).

We performed the following pre-processing steps:

1. The following source domains were selected: Laboratory measurements (lm), adverse event information (ae), vital signs (vs), concomitant medication (cm), and demographic information (dm).

2. Information on the domains were formulated to longitudinal samples $Y$ comprising of (lb, vs), and auxiliary covariates $X$ comprising of (dm, ae, cm). Only measurement timepoints in ae and cm with known start and end time were considered.

3. Timepoints with less than 70% of information in $Y$ were excluded.

4. The 10 most common ae and cm in X were considered.

5. Patients with a trajectory of less than 5 observations were excluded.

6. Each feature in $Y$ was standardised to have zero mean and unit variance.

The final pre-processed dataset used for modelling is composed of 480 subjects with $N = 6605$ total observations, $\boldsymbol{y} \in \mathbb{R}^{30}$. and $\boldsymbol{x} \in \mathbb{R}^{24}$. Following one-hot encoding of the patient ids, we have $\boldsymbol{x} \in \mathbb{R}^{503}$.

The approach to constructing the train-validation-test sets is similar to that for Health MNIST. For the test set, we consider the sequence of visits following the $5^{th}$ visit (inclusive) of 100 patients. The first 4 visits of these aforementioned patients are included in the train set. The remainder of the dataset is then split amongst the train and validation sets.

## L   Identifiability Experiments

### L.1   Clinical Trial Data (Project Data Sphere)

The actual architectures used for LMM-VAE and the vanilla VAE can be found in Table 13.

For experiments regarding the VAE and LMM-VAE, we rely on the Adam optimizer Kingma & Ba (2015) and a learning rate of 0.001. We also use an exponential learning rate scheduler with a step size of 500. We monitor the loss on the validation set and employ a strategy similar in spirit to early stopping, where we save the weights of the model with the optimal validation loss. LMM-VAE was allowed to run for a maximum of 2000 epochs. We also define $\sigma_z = 1$.

For all experiments, we report the mean and standard deviation obtained across five runs.

### L.2   Health MNIST

The actual architectures used for LMM-VAE and the vanilla VAE can be found in Table 14.

For experiments regarding the VAE and LMM-VAE, we use the Adam optimizer Kingma & Ba (2015) and a learning rate of 0.001. We also use an exponential learning rate scheduler with a step size of 500. We monitor the loss on the validation set and employ a strategy similar in spirit to early stopping, where we save the weights of the model with the optimal validation loss. LMM-VAE was allowed to run for a maximum of 2000 epochs. We also define $\sigma_z = 1$.

For all experiments, we report the mean and standard deviation obtained across five runs.

## M   Model Architectures

This section outlines the different model architectures used for the different experiments. Table 10 contains the neural network architecture used in the Health MNIST experiments, which follows Ramchandran et al. (2021). Table 11 describes the architecture used for the Rotating MNIST experiments. This architecture is almost identical to that reported in Casale et al. (2018), apart from a final activation layer of sigmoid instead of the ELU to suit the modified pre-processing pipeline. Table 12 details the architecture used for the Physionet Challenge experiments. Finally, Tables 13 and 14 for the experiments on identifiability, where the decoder is composed of fully connected layers and ELU/Sigmoid activation.

### M.1   Identifiability Experiments

The architecture used for the RCT dataset is described in Table 13, while that for Health MNIST is in Table 14.

Table 10: Neural Network Architecture used for the model in the Health MNIST experiments.

|  | Hyperparameter | Value |
| --- | --- | --- |
| Inference Network | Dimensionality of input | $36 \times 36$ |
|  | Number of convolution layers | 2 |
|  | Kernel size | $3 \times 3$ |
|  | Stride | 2 |
|  | Pooling | Max Pooling |
|  | Pooling kernel size | $2 \times 2$ |
|  | Pooling stride | 2 |
|  | Number of feedforward layers | 2 |
|  | Width of feedforward layers | 300, 30 |
|  | Dimensionality of latent space | L |
|  | Activation function of layers | ReLU |
| Generative Network | Dimensionality of input | L |
|  | Number of transposed convolution layers | 2 |
|  | Kernel size | $4 \times 4$ |
|  | Stride | 2 |
|  | Number of feedforward layers | 2 |
|  | Width of feedforward layer | 30, 300 |
|  | Activation function of layers | ReLU, Sigmoid |

Table 11: Neural Network Architecture used for the model in the Rotating MNIST experiments.

|  | Hyperparameter | Value |
| --- | --- | --- |
| Inference Network | Dimensionality of input | $28 \times 28$ |
|  | Number of convolution layers | 3 |
|  | Kernel size | $3 \times 3$ |
|  | Stride | 2 |
|  | Number of feedforward layers | 1 |
|  | Width of feedforward layers | 128 |
|  | Dimensionality of latent space | L |
|  | Activation function of layers | ELU |
| Generative Network | Dimensionality of input | L |
|  | Number of convolution layers + Upsampling | 3 |
|  | Kernel size | $3 \times 3$ |
|  | Stride | 1 |
|  | Number of feedforward layers | 1 |
|  | Width of feedforward layers | 128 |
|  | Activation function of layers | ELU, Sigmoid |

Table 12: Neural Network Architecture used for the model in the Physionet Challenge experiments.

|  | Hyperparameter | Value |
| --- | --- | --- |
| Inference Network | Dimensionality of input | 37 |
|  | Number of feedforward layers | 2 |
|  | Number of hidden units per layer | 64, 32 |
|  | Dimensionality of latent space | L |
|  | Activation function of layers | ELU |
| Generative Network | Dimensionality of input | L |
|  | Number of feedforward layers | 2 |
|  | Dimensionality of input | 32, 64 |
|  | Activation function of layers | ELU |

Table 13: Neural Network Architecture used for the model in Project Datasphere's experiments.

|  | Hyperparameter | Value |
|---|---|---|
| Inference Network | Dimensionality of input | 30 |
|  | Number of feedforward layers | 2 |
|  | Number of hidden units per layer | 25, 23 |
|  | Dimensionality of latent space | L |
|  | Activation function of layers | ELU |
| Generative Network | Dimensionality of input | L |
|  | Number of feedforward layers | 2 |
|  | Dimensionality of input | 23, 25 |
|  | Activation function of layers | ELU |

Table 14: Neural Network Architecture used for the model in Health MNIST's experiments.

|  | Hyperparameter | Value |
|---|---|---|
| Inference Network | Dimensionality of input | $36 \times 36$ |
|  | Number of convolution layers | 2 |
|  | Kernel size | $3 \times 3$ |
|  | Stride | 2 |
|  | Pooling | Max Pooling |
|  | Pooling kernel size | $2 \times 2$ |
|  | Pooling stride | 2 |
|  | Number of feedforward layers | 2 |
|  | Width of feedforward layers | 300, 30 |
|  | Dimensionality of latent space | L |
|  | Activation function of layers | ReLU |
| Generative Network | Dimensionality of input | L |
|  | Number of feedforward layers | 4 |
|  | Dimensionality of input | 30, 300, 1000, 1200 |
|  | Activation function of layers | ELU, Sigmoid |

