# OpenReview forum: "Latent mixed-effect models for high-dimensional longitudinal data"
_TMLR — Accepted by TMLR_

### Review · Reviewer_xWFG · 2025-02-14

**Summary Of Contributions:**

The authors deal with an important gap in the literature - how to handle non-i.i.d. / clustered / longitudinal data whilst making minimal assumptions regarding the functional form. They propose a combination of the LMM with variational inference. The literature on linear mixed models is extensive, and as described by the authors, has wide applicability given the importance of accounting for the impact on variance associated with samples arising from shared processes (e.g. patients from the same hospital, measures from the same patient). Their proposed methods (oLMM-VAE and sLMM-VAE) are compared for their efficacy at missing value imputation, forecasting, and interpolation.

**Audience:**

Yes

**Broader Impact Concerns:**

None.

**Claims And Evidence:**

Yes

**Requested Changes:**

The requested changes can be inferred readily from my list of comments above.

**Strengths And Weaknesses:**

On the whole the paper is easy to read and follow, and I think the contribution is both valid and valuable to the community. That said, I have some comments:

So in the interests of transparency, some of my work falls at the intersection of machine learning, statistics, and importantly, empirical research in the medical, health, and social domains. As such, I am quite familiar with the need for models which account for the hierarchical nature of data. One of the key motivations for this is to account for the shared variance for (e.g.) multiple measures from the same patient. If one fails to sufficiently deal with this, the associated statistical inference is bias (underestimated standard errors). Whilst there is also a more general case that accounting for nested structure helps the model more closely align with the data generating process (DGP), and therefore the model is more predictive, the case for mitigating bias for statistical inference is extremely important for drawing conclusions about e.g. the efficacy of treatments. With this in mind, I have the following comments:

1) The paper could benefit from a strong/simple motivating example which is carried through the methodology. This is particularly important in my view, because one of the key audiences for this paper could include people working in the applied domains, and tying the approach to a more conventional example would be particularly helpful given that the authors use a formalism which is not quite the same as is usually seen in LMM literature (i.e. y_{i,j} =… where i is the individual and j is the cluster/timepoint). I would recommend the authors bring this example back up briefly at each stage of the exposition, to tie each proposition back to something tangible.
2) Relatedly, Figure 2 is fine, but doesn’t really help one represent the problem the authors are trying to deal with - there is no representation of the nested dimension (e.g. time) in this figure. Perhaps it would be worth including a second figure here (or modifying the current figure), showing more explicitly how the DGP contains a hierarchical structure. In my view, this wouldn’t be unusual, and honestly could even be argued to be one of the main purposes of PGMs.
3) I’m curious/surprised about the choice of evaluations. In some ways, I feel like there was a strange mixture of conventional computer vision with health/medical (‘Time from disease event’ on top of some rotating MNIST images was particularly artificial). I think the image example is nice but should either (a) be replaced with an image example which is more relevant (medical imaging?) and/or (b) focus more on the clinical trial and physio examples.
4) Relatedly, the evaluation metrics used make sense, but would there be a way to assess the extent to which key statistical parameters are better estimated with this method? As mentioned, biased standard errors are one of, if not <the> most important motivation for mixed modelling. If I were an empirical researcher with time series data, I’d want to use the authors’ method because of the flexible function form, but be a bit lost as to whether it actually does the job in terms of facilitating statistical inference on <simple> tasks which I might otherwise fall back on a conventional LMM analysis. Relatedly, where is the benchmark here for simple/linear tasks - if I use LMM-VAE instead of, say, the lmer package in R, how do they compare?
5)  Could the authors comment on the Table 6 MCC results - why do the authors think the VAE is so close to the oLMM-VAE?


Minor:

Clarify that LMMs are also known as, hierarchical linear models, multilevel models, mixed effects models (fixed and random effects).
Add equation numbers to equations on page 4.

---

> ### Author Response · Authors · 2025-02-28
> **Official Comment by Authors**
>
> #### 1. A running example
>
> Thank you for your suggestion. We have added an example to the revised manuscript illustrating how the notation for LMM-VAE is derived from the LMM formulation. This can be found in Section 3, under "Linear Models."
>
>
> #### 2. Figure 2
>
> Thank you for your suggestion. We have added a supplementary plate diagram to Appendix B that reflects the hierarchical nature of our model.
>
>
> #### 3. Choice of evaluations
>
> The evaluation datasets were carefully selected to 1) reflect characteristics commonly found in modern high-dimensional longitudinal datasets, and 2) align with prior research on VAEs that incorporate covariate information into their modelling frameworks, such as GPP-VAE, SVGP-VAE, and LVAE. Notably, previous studies in this area [1,2,3], which focus on augmenting the VAE prior to enhance predictive performance, have relied on variants of the MNIST dataset for evaluation.
>
> Additionally, we included the PhysioNet dataset in our experiments to assess LMM-VAE in a realistic healthcare setting, given its recognition as a widely used benchmark [4]. Another real-world dataset that demonstrates LMM-VAE's identifiability is the Clinical Trial dataset from Project Data Sphere, which is detailed in the Appendix. With these evaluations, we believe our approach has been rigorously tested across synthetic and real-world datasets, with further evaluations on additional clinical, medical, or bioimaging datasets left for future work.
>
>
>
> #### 4. Estimation of statistical parameters
>
> Our proposed model is not intended to replace classical LMMs for linear modelling tasks but rather to extend these methods to more complex, high-dimensional settings with potentially highly non-linear mappings from the latent variables to observations. Consequently, most of our benchmarks focus on higher-dimensional setups, where LMM-VAE is designed to operate.
>
> As a side note, deep generative models are generally less statistically calibrated compared to classical LMMs. However, since the ELBO objective includes the log-likelihood term for modelling the data, the model theoretically has the potential to be calibrated if the global optimum is achieved.
>
>
> #### 5. MCC results
>
> The MCC score serves as an empirical measure of identifiability; however, we note that achieving high MCC scores can be challenging, as identifiability is fully realized only asymptotically. We hypothesize that the current dataset may not be large enough to produce empirical differences in MCC scores.
>
>
> [1] Francesco Paolo Casale et al. Gaussian Process Prior Variational Autoencoder. In: Neural Information Processing Systems. 2018.
>
> [2] Metod Jazbec et al. Scalable Gaussian Process Variational Autoencoders. In: International Conference on Artificial Intelligence and Statistics. 2021.
>
> [3] Siddharth Ramchandran et al. Longitudinal Variational Autoencoder. In: International Conference on Artificial Intelligence and Statistics. 2021.
>
> [4] Moody GB, Mark RG, Goldberger AL. PhysioNet: physiologic signals, time series and related open source software for basic, clinical, and applied research. Annu Int Conf IEEE Eng Med Biol Soc. 2011.

---

> > ### Comment · Reviewer_xWFG · 2025-03-05
> > **Response to rebuttal**
> >
> > Thanks very much to the authors for their response, and for receiving the comments in the same spirit in which they were intended (i.e. constructively!).
> >
> > Notwithstanding any open discussion points with other reviewers, I think for me the biggest 'gap' still remaining relates to my point (4).
> >
> > I understand LMM-VAE is not intended to replace the LMM class (for multiple reasons), I nonetheless believe that the typical LMM serves as a both a useful baseline <and> a useful target. It is a baseline in the sense that it will perform poorly on data from non-linear DGPs (and probably not remotely function on image data),  and it is a target in terms of the statistical parameters which is can recover for data from linear, low-D / 'tabular' DGPs (i.e. under optimum conditions for LMMs).
> >
> > The compelling use case for me is as follows: I have tabular, repeated-measures data (e.g. medical/physio data), and I want to estimate the effect of an intervention. Assume that I <fear> that the relationships in my data are non-linear (but might actually be 'sufficiently' linear in reality). What I'd like to do is to fit both LMM and LMM-VAE to the data and compare the results (e.g. by looking at the distribution of the residuals for each model). I'd want to know that LMM-VAE is a <viable> approach in this instance, and that they even provide similar/close estimates under equivalent conditions.
> >
> > I could see various outcomes: Either (a) LMM-VAE, in practice, does not perform well on small, close-to-linear datasets, and that LMMs are much more performant, or (b) LMM-VAE might be less data-efficient, but provides comparable results to LMM on small datasets.  In the case of (a), it would be important to clearly define the scope of potential usage for LMM-VAE (or equivalently to define what it <can't> do). In other words, if (a) is the case, I'd want to rule out LMM-VAE for typical low-D, close-to-linear modeling tasks, if they really can't perform. And of course there are various dimensions of 'performance' that are relevant for this comparison - the data efficiency one first comes to mind, but I'm not sure this is totally interesting (we know that LMM-VAE will be less data-efficient) - the ability to converge on a useful/close-to-equivalent solution is perhaps more interesting.
> >
> > I think this kind of comparison is missing. Given that LMMs are one of the most obivous comparators/benchmarks, I think more should be done to establish their relative performance and domains of application.
> >
> > That said, I once again think that the method is a good contribution.

---

> > > ### Author Response · Authors · 2025-03-12
> > > **Official Response by Authors**
> > >
> > > Thank you for this follow up question and the description of the precise scenario you are interested in.
> > >
> > > Most LMM setups and libraries we are aware of, e.g., `lmer` in R, or `statsmodel` in python assume low-dimensional/scalar responses. While we can create such a low-dimensional/scalar response experiment as well, it would in our opinion remove most of the model's motivation.
> > >
> > > It would leave the non-linear flexibility as the remaining feature where the decoder becomes a regular non-linear mapping.
> > >
> > >   A proper experiment would in that case compare
> > >
> > > 1. a linear mixed model which assumes a linear relation between covariates and response;
> > > 2. a nonlinear mixed model which assumes a fixed, usually known, non-linear mapping;
> > > 3. an ablation of LMM-VAE of the form $y = f(Ax) + \epsilon$, i.e., where $f$ replaces the decoder;
> > > 4. a full LMM-VAE which incorporates the a potentially regularizing guidance of an encoder.
> > >
> > > Regarding our proposed model (item #4 above), as the auto-encoder would map a low-dimensional/scalar value to a higher dimensional latent variable, there will most likely not be a regularizing effect and on the contrary additional overfitting potential due to the additional complexity. This is not a setting where auto-encoding variational Bayes is typically applied or designed for.
> > > This leaves the experiment to comparing 1.-3., and thus the question: can a parametric neural net serve as a variation of classical non/linear mixed models? Ideally it would be able to vary between inferring a linear relationship, recovering a known non-linear one, and learning a fully unknown mapping.
> > >
> > > While we consider this to be an interesting research question in itself, it seems to be somewhat out of scope and would deserve a full paper on its own in our opinion.

---

> > > > ### Comment · Reviewer_xWFG · 2025-03-12
> > > > **Response to Point about LMM Baseline**
> > > >
> > > > Thanks once again to the authors for their considered responses!
> > > >
> > > > I tend to agree with most of what they authors say, and indeed, I don't mean to suggest that LMM-VAEs replace LMMs. However, I do consider the linear, low-dimensional setting to fall entirely within the class of models which can be represented using LMM-VAEs, <even if> LMMs are preferred and more data-efficient in such a setting.
> > > >
> > > > To this extent, whilst an extensive evaluation is (I agree) beyond the scope of the paper, it seems totally reasonable to check that LMM-VAEs converge (at least in the large-data setting) to a solution which is equivalent to that of an LMM. If the LMM-VAE fails in such a simple setting, why would I trust it to work in more complex settings? Of course, if I 'only' want a predictive model which generates reasonable outputs, perhaps I wouldn't be worried about statistical parameter estimation tasks. But statistical estimation tasks are extremely important for applications which are sensitive (e.g. medical outcomes with longitudinal / repeated-measures data - and I believe this <is> within the motivational scope of LMM-VAEs). I believe the authors' argument (if I paraphrase) that 'the low-dim linear setting is not the principal/motivational setting for the LMM-VAE' is insufficient to justify the absence of such an evaluation, even if it is true as an independent statement.
> > > >
> > > > For me, such a baseline comparison would represent one of the first 'sanity checks' that my model fulfils the most basic requirements within the class of requirements it promises to fulfil. And if it does not... I'd be concerned as to why.

---

> > > > > ### Author Response · Authors · 2025-03-13
> > > > > **Official Response by Authors**
> > > > >
> > > > > Thank you for the continued discussion.
> > > > >
> > > > > We would like to clarify the setup to make sure we fully understand your desired experiment.
> > > > >
> > > > > If we understand correctly, you’re looking for a setup with a one-dimensional response variable $y$. In this scenario, an auto-encoder does not make any sense and the natural choice is to revert to a deterministic identity mapping from $\mathcal{Z} \to \mathcal{Y}$, i.e., $f_\theta: = \text{id}(\cdot)$, $p(y|z,\theta) = \delta(y=z)$, where $\delta$ is the $\delta$-distribution, such that $y = z$ and the LMM-VAE collapses to a regular LMM---to be precise a Bayesian LMM---and, therefore, performs identically.
> > > > >
> > > > > If the auto-encoder structure is kept, maybe even with $\dim(\mathcal{Z}) \geq \dim(\mathcal{Y})$, the theoretical optimum is again for the model to learn an identity mapping collapsing the normal likelihood and variational posterior to delta distributions up to numerical precision and reverting to a regular Bayesian LMM.
> > > > > In this case, worse empirical performance would simply indicate suboptimal optimization, due to gradient noise, suboptimal learning rate convergence, and numerical imprecisions as the inferred variance can never become zero in practice.
> > > > >
> > > > > Are we correct in understanding that the request is to assess whether our current implementation can achieve the theoretical optimum?

---

### Review · Reviewer_ZB6M · 2025-02-14

**Summary Of Contributions:**

Motivated by the problem of poor scalability of VAEs to real-world longitudinal datasets, that usually contain non-linear effects, time-varying covariates and missing values, the authors introduce the linear mixed model VAE (LMM-VAE). Thus, the driving idea is to make improvements in terms of interpretability, identifiability and last, make the model available to the introduction of basis functions (i.e, of periodic nature), such that the model can capture certain types of structures in the temporal/longitudinal data. In practice, the model is a VAE, where one assumes the latent variable is a noisy linear projection of some additional auxiliary covariate $X$, given observed outputs $Y$. The experiments reflect some improvement w.r.t. more complex and well-established SOTA methods, mostly in the context of GP-VAEs and sequantially-structured VAEs.

**Audience:**

Yes

**Claims And Evidence:**

No

**Requested Changes:**

I do believe the work need a thorough revision, and I am certainly concerned about two principal issues (I would like to see this solved somehow):

- *1)* Is there a connection with PPCA? Are LMM-VAEs actually something similar to them? Could authors review that side of the literature appropriately and give a bit more of clarity and wide vision on that side?

- *2)* Given the experiments considered, could the authors make a thorough study of the behaviour of the method in the context of latent space dimensionality, structure and number of parameters involved?

**Strengths And Weaknesses:**

**Strengths:**

The paper tackles a well-known problem in the context of VAEs and representation learning, where usually temporal structure of longitudinal is considered as a second priority (in the sense that usually the focus has been so long centered on reconstruction, OoD and uncertainty quantification). In that sense, I find the paper interesting and kind of in a good direction when reviewing previous SOTA works and problems. One of the strengths of the work comes from the fact of considering longitudinal data that is multimodal (either regression or classification problems, that could include categorical variables for instance). However I do believe that the paper do not develop well this side of the topics and do not link itself to some good literature on that sort of problems written during the last years. A good style of writing and in general, clarity, are there — in that sense I did not detect any problem on understanding what the authors are proposing and what they want to do.

**Weaknesses:**

To me, the most surprising aspect while reading the paper is the lack of reference or mention to principal component analysis (PCA), its probabilistic version (PPCA), the GP adaptation of such vast literature, super well-known as Gaussian process latent variable models (GPLVMs) and dozens or other variants that have been around the community in the last years under names like GP-decoders or similar titles.

In that regard, it is hard for me to believe that VAE models updated with sorts of PPCA structure have not been studied since their apparition +10y ago. In fact, the linear consideration of both encoders and decoders have been around since the beginning for the understanding and analysis of such generative models.

Even if we assume that the LMM-VAE is not related to PPCA (I’m very sure it is), its connection with GPs and basis functions is trivial… or at least, there is not novelty there for anyone with a little background on probabilistic linear models, GPs and the weight/function-space view duality of the previous ones.

Last but not least, the experiments are certainly limited and show important flaws. To name two of them: I do not entirely believe the results or conclusions drawn from the tables and figures that show improvements of one order of magnitude wrt well-established and rigorous methods in the side of longitudinal-structured VAEs. Why I say this? Well, the question that quickly comes to mind is if that performance with dim($z$)$=32$, dim($z$)$=30$ or dim($z$)$=16$ and zero variance on the MSE accuracy for the Rotated/Health versions of the MNIST dataset really responds to a great method, or is just an oddly tuned experiment that worked well for such dimensions. I kindly ask to have a bit of more rigour on these sort of experiments, particularly to show readers the complete and clear performance of the method for different latent space structures and dimensions. The computational cost would be also a topic to consider here, but lets let it out of the discussion for now..

---

> ### Author Response · Authors · 2025-02-28
> **Official Comment by Authors**
>
> #### 1. Positioning of work
>
> Thank you for your question. We believe there might have been some misunderstanding with respect to our model design, and we will do our best to clarify any misunderstandings regarding its focus.
> | Model Class | Auxiliary Covariates | Latent Prior (Z)                | Decoder               |
> |-------------|----------------------|---------------------------------|-----------------------|
> | PPCA        | No                   | $$\mathcal{N}(0, I)$$          | Linear                |
> | BGPLVM      | No                   | $$\mathcal{N}(0, I)$$          | Non-linear (GP)       |
> | VAE         | No                   | $$\mathcal{N}(0, I)$$          | Non-Linear            |
> | GP-VAEs     | Yes                  | $$\mathcal{N}(0, K)$$          | Non-Linear            |
> | LMM-VAE     | Yes                  | $$\mathcal{N}(Ax, \sigma^2 I)$$| Non-Linear            |
>
>
>
> Variational Autoencoders (VAEs) are neural architectures that consist of a generative model that is learned using amortized variational inference. The generative model is defined by a latent variable
> $z$ drawn from a prior distribution $p(z)$, and an observed variable $y$ generated from the conditional $p(y|z)$ distribution (also called decoder). When the decoder is parametrized as a linear function, the generative model reduces to PPCA. Alternatively, if $p(y|z)$ is assigned a GP prior, the generative model becomes a Bayesian GPLVM.
>
> Our work, however, focuses on conditional VAEs, where the latent variable $z$ is conditioned on auxiliary covariates $x$, expressed as $p(z|x)$. This framework allows the model to capture the influence of
> $x$ on the latent representation $z$, thereby enriching the generative process of $y$ through $p(y|z)$.
>
> The key contribution of our approach lies in the enhancement of the latent space representation, enabling us to effectively model correlations in longitudinal data. In this context, we believe that neither PPCA not BGPLVM are the most relevant comparison, as our focus is on structured priors that account for dependencies in the data rather than purely linear latent-variable models or GPLVMs.

---

> > ### Author Response · Authors · 2025-02-28
> > **Official Comment by Authors**
> >
> > > Last but not least, the experiments are certainly limited [...] the tables and figures that show improvements of one order of magnitude wrt well-established and rigorous methods in the side of longitudinal-structured VAEs.
> >
> > We believe this statement misrepresents our evaluation, and we aim to clarify any misunderstandings in this response.
> >
> > With the exception of one experimental setup computing the imputation MSE on Health MNIST, our evaluations of LMM-VAE neither report zero inter-seed variance nor an order of magnitude improvement over baseline methods. For synthetic datasets, we strive to follow the experimental setups outlined in prior work, including the latent dimensionality [1,2,3]. To ensure additional rigor, we conduct an ablation study on LMM-VAE's predictive performance with respect to the latent dimension for both Health MNIST and Project Data Sphere, which are detailed in Appendices F1 and E1, respectively. Furthermore, our results on PhysioNet show that while our method is competitive, it does not outperform the baselines, which we believe counters the characterization of an "oddly tuned experiment".
> >
> >
> > [1] Francesco Paolo Casale et al. Gaussian Process Prior Variational Autoencoder. In: Neural Information Processing Systems. 2018.
> >
> > [2] Metod Jazbec et al. Scalable Gaussian Process Variational Autoencoders. In: International Conference on Artificial Intelligence and Statistics. 2021.
> >
> > [3] Siddharth Ramchandran et al. Longitudinal Variational Autoencoder. In: International Conference on Artificial Intelligence and Statistics. 2021.

---

### Review · Reviewer_hND8 · 2025-02-17

**Summary Of Contributions:**

We would often like to model longitudinal data in order to impute missing values or predict new values, for example. However, such data might be very high dimensional, and so it is more effective to instead model the data through a low dimensional latent space—typically achieved with VAEs. In order for VAEs to be able to model the correlations between samples within the same task (i.e. the state of the features representing a patient's health at different times, where the time feature is termed an *auxiliary feature*), the priors over the elements of the latent variable $z$ are endowed with some way to condition on the auxiliary variable. In previous work, this has been done by using GP priors. Since GPs are a kernel method and therefore suffer when the number of auxiliary features is high, the authors propose to use linear mixed models instead, which are more scalable in the number of auxiliary features. Importantly, their approach maintains scalability in the number of datapoints too, and is easy to train unlike existing sparse GP VAE approaches. They present two variants of their model, and propose two objective functions. They also open the door to non-linearising their linear models via the kernel trick. The proposed methods exhibit solid performance compared to the benchmarks across all experiments.

**Audience:**

Yes

**Broader Impact Concerns:**

N/A.

**Claims And Evidence:**

Yes

**Requested Changes:**

I use this section to request changes to the manuscript as well as outline my concerns in general.

1. From the paper, I do not feel like I understand what the *random* features are or how they work in the model. It would be great if a clearer explanation between the role of these vs. the *shared* features could be given, perhaps by way of more/better examples.
2. There are certain passages that are either in an appendix and need to be brought into the main text, or vice versa. I outline what I find to be the key offenders of this as follows:
    - The ELBO derivation at the bottom of page 5 should be in an appendix, with only the key result remaining in the main text.
    - An intuitive explanation of the sLMM-VAE encoder that sits in the main text would be great.
    - Section 5 (Identifiability) is meaningless to the reader without a convincing motivation for an identifiable model in the first place---bring (at least part of) the subsection on 'Motivation' of appendix E into the main text.
3. Non-linearising the LMM with a fixed number of basis functions is, in spirit, the same thing as using a sparse GP (with, here, the advantage that inference is performed in parameter space). However, does this not mean you run into the same *curse-of-dimensionality* problems that GPs/kernel methods have for large numbers of features---does the kernel trick here not limit the scalability of the LMM-VAE in the number of auxiliary variables (the very problem you seek to address)? Perhaps I have misunderstood something important here.
4. In the future work section, you mention using NNs for prior means. Would this not destroy the nice interpretability that the (identifiable and linear) LMM-VAE enjoys with respect to the auxiliary variables?
5. You explore the effect of varying the number of basis functions in the kernelised LMM-VAE a little bit, but never compare to varying the number of inducing points in the SGP-VAE. Further, I cannot find anywhere in the paper where the number of inducing points in the SGP-VAE is reported. It would be good to ensure the degree of sparsity is similar in order to be confident that the comparison is indeed a fair one.
6. Minor issue, but appendix D has the wrong title.

**Strengths And Weaknesses:**

**Strengths**
- Well motivated problem.
- A nice related work section and a generally solid introduction to the area.
- Strong experimental results.

**Weaknesses**
- I find the exposition unclear at times.
- More discussion/comparison with SGP-VAE is needed.

---

> ### Author Response · Authors · 2025-02-28
> **Official Comment by Authors**
>
> #### 1. Random effects vs Shared effects
>
> Thank you for your comments. Random and fixed effects are common terms in the LMM literature, referring to instance-specific (or group/cluster-specific) effects and effects that are consistent across all observations, respectively. We have clarified this distinction in the uploaded version of the manuscript and will elaborate further below.
>
> As an example, if the covariate set consists of three features (excluding interaction terms) -categorical ID, age, and time — we would categorize the variables into two sets: one associated with random effects, and another associated with fixed effects:
>
> Random effects: Categorical ID
> - Each ID (representing a distinct individual) has a unique offset, capturing group-level variation.
>
> Shared effects: age, time
> - The effect of age and time is assumed to be consistent across all observations.
>
> Modelling both random and fixed effects allow us to yield more structured representations of the latent variables. This modelling choice allows us to capture temporal dependencies, account for group-level variations, as well as maintain generalizability across observations.
>
>
> 2. > There are certain passages that are either in an appendix and need to be brought into the main text, or vice versa.
>
> Thank you for your feedback! We've incorporated your suggestions into the revised version of our manuscript.

---

> > ### Author Response · Authors · 2025-02-28
> > **Official Comment by Authors**
> >
> > #### 3. LMM-VAE's Complexity
> >
> > Thank you for your question.
> >
> > We would like to emphasize that LMM-VAE can model an arbitrary number of covariates, a capability that many baseline methods struggle to achieve. While we acknowledge that the computational complexity increases with the number of covariates, LMM-VAE is still able to accommodate all of them, whereas many baselines are limited to considering only a subset due to their modelling assumptions. We would be happy to engage in further discussion and provide additional details if needed.
> >
> >
> > #### 4. Alternative NN priors
> >
> > We agree that not every NN prior will yield the benefits observed in LMM-VAE.
> > In practice it is up to the user to specify the level of interpretability they require for their respective experimental setup.
> >
> > Exploring this further remains an avenue for future work.
> >
> > #### 5. Number of inducing points
> >
> > Thank you for your comment. To ensure a fair comparison with baseline methods, we utilized the default number of inducing points as specified in their official GitHub implementations. These reference implementations were designed for datasets similar to those in our study, and we believe that the hyperparameters used in these implementations yield results that accurately reflect their capabilities. While maintaining equivalent sparsity across methods would be ideal, implementing this uniformly presents challenges due to fundamental differences in how each method introduces sparsity. Exploring performance variations across different sparsity levels could be valuable future work. However, our primary contribution—and the focus of our evaluations—is the enabling of scalable modeling with an arbitrary number of covariates, a capability that many baseline methods struggle to accommodate.

---

> > > ### Author Response · Authors · 2025-02-28
> > > **Official Comment by Authors**
> > >
> > > > More discussion/comparison with SGP-VAE is needed.
> > >
> > >
> > > Thank you for your comment. We would like to take this opportunity to discuss the complexity of LMM-VAE in comparison to the baseline models.
> > >
> > > For this discussion, we focus on the computational complexity per iteration of learning the Variational Autoencoder (VAE) prior, assuming a fixed latent dimension. Since the encoder and decoder architectures are consistent across the baseline models, they are not a focus of this analysis.
> > >
> > > ##### GP prior:
> > >
> > > The computational complexity of vanilla GPs is $O(N^3)$, where N refers to the number of samples. This cubic complexity limits the scalability and application of GPs to large datasets. To address this issue, various approximation methods have been developed. Common strategies include Taylor approximations (e.g., GPP-VAE) and the use of inducing points (e.g., SVGP-VAE and LVAE).
> > >
> > > In the case of GPP-VAE, the complexity of learning its prior is $O(NH^2)$, where $H$ is the dimension of the low rank matrix used to approximate the GP covariance [1]. While this represents an improvement over the original  $O(N^3)$ complexity, this model class is still not optimal for longitudinal datasets, as it does not easily scale to a large number of covariates.
> > >
> > > Next, we discuss the complexity associated with learning the GP prior using inducing points. Of particular interest is LVAE, which accommodates an arbitrary number of covariates in the prior. To make a fair comparison, we assume a set of $M$ inducing points. With inducing points, the complexity for the SVGP-VAE [2] is reduced to $O(bM^2 + M^3)$. For LVAE [3], the complexity is  $O(bM^2 + \sum_{p \in B} n_p^3)$, where $n_p$ is the number of observations associated with instance $p$ in minibatch $B$ of size $b$. Furthermore, since some of the covariates are typically categorical, such as gender, inducing point locations cannot be optimized using the existing gradient-based methods, thus further decreasing performance of GP-based models.
> > >
> > > While SVGP-VAE and LVAE significantly reduce the computational burden compared to standard GPs, the number of inducing points required for effective modelling often scales with the complexity of the problem. As the overall complexity is still cubic with respect to $M$, these methods can still incur high computational costs. In practical applications, such as in medical domains, this scaling issue can pose a major bottleneck, potentially making the model computationally prohibitive depending on the size and complexity of the dataset.
> > >
> > >
> > > ##### LMM prior:
> > >
> > > LMM-VAE does not depend on inducing points; instead, it learns the model through global parameterization. The computational overhead associated with optimizing the LMM prior, driven by stochastic gradient descent (SGD), is $O(BF)$, where $F$ represents the total number of parameters. Although the number of parameters for effective modelling may increase with the complexity of the problem, this growth remains linear with respect to the number of parameters. This makes LMM-VAE more scalable in comparison to approaches that rely on inducing points.
> > >
> > >
> > > [1] Francesco Paolo Casale et al. Gaussian Process Prior Variational Autoencoder. In: Neural Information Processing Systems. 2018.
> > >
> > > [2] Metod Jazbec et al. Scalable Gaussian Process Variational Autoencoders. In: International Conference on Artificial Intelligence and Statistics. 2021.
> > >
> > > [3] Siddharth Ramchandran et al. Longitudinal Variational Autoencoder. In: International Conference on Artificial Intelligence and Statistics. 2021.

---

### Comment · Action_Editor_BiJR · 2025-02-17
**Reviews arrived**

The reviews have arrived. I have read them carefully, and, while I don't want to repeat them, I would like offer a quick summary and emphasize a few priorities for the revisions:
* **Clarity in presentation:** Overall, the reviews were largely positive on the contribution but still some open questions persist. I think all of them are addressable and I usher the authors to carefully respond to each (and to make meaningful revisions to the paper). In general, 2 out of 3 reviewers had some negative perceptions regarding clarity but also made some helpful suggestions (e.g., adding an example).
* **Embedding in the literature:** there are some good suggestions on how to make the link -- and differences -- to the literature clearer.
* **Experiments:** The experiments were overall perceived as good, but there were specific questions regarding settings and baselines. I would strongly advise the authors to follow these suggestions closely and carefully respond to them.

---

> ### Author Response · Authors · 2025-02-28
> **Global Response**
>
> We sincerely thank the reviewers for their insightful comments and thoughtful questions.
>
> We appreciate the reviewers' recognition of the strengths of our work and the significance of the problem we are addressing. Specifically, they acknowledged that our problem is well-motivated (hND8), our contributions are valuable (xWFG), and our experimental results are robust (hND8).
>
> In response, we have provided detailed replies to each reviewer’s comment, focusing on addressing concerns regarding the experimental setup, clarifying our contributions, reinforcing the positioning of our work within the existing literature, and resolving any potential misunderstandings. Additionally, we have uploaded a revised version of the manuscript, with deletions marked in red and new additions in blue. The revisions include:
>
> - A strengthened related works section (Section 2).
> - An example illustrating the connection between the formalism of LMMs and the approach of LMM-VAE (Section 3).
> - A supplementary plate model highlighting the hierarchical nature of the LMM-VAE framework (Appendix B).
> - A detailed description of the SLMM-VAE encoder (Section 4.3).
> - A clearer motivation for identifiability (Section 5).
>
> We would be happy to engage in further discussions and provide additional details if needed.

---

### Comment · Action_Editor_BiJR · 2025-03-05
**Further suggestions**

I am getting back the votes from the reviewers. I do hear positive comments but the reviewers also point to things that could be further improved:

* Can the authors further improve their work to offer 1) a more rigorous comparison with the SGP-VAE, and explain 2) why the kernelised LMM-VAE remains robust against the curse of dimensionality when the number of covariates is high

---

> ### Author Response · Authors · 2025-03-12
> **Official Comment by Authors**
>
> Thank you for the update.
>
> @Reviewers: In case there are additional questions that you would like to discuss, please don't hesitate to contact us directly as well.
>
>
> > 1) a more rigorous comparison with SGP-VAE
>
> Thank you for your suggestion.
>
> Jazbec et al. (2021)[1] tend to use between 5 and 32 inducing points depending on the experiment, for varying data set sizes, e.g., their rotating MNIST-based experiment has $N=4050$ observations and uses 32 points. Our adaptation has $N=4400$ training observations and we provide SVGP-VAE with two inducing points for each of the 16 angles such that we end up with 32 inducing points, i.e., the optimal set-up reported in Jazbec et al. (2021). (See, e.g., the discussion in Appendix C.4 and Figure C.3 therein.).
>
> We also include an additional comparison between SVGP-VAE and LMM-VAE below, which we conduct on the rotating MNIST dataset.
>
>
>
> **Improving performance with inducing points**
>
> The approximation quality of LMM-VAE improves as the number of basis functions increases. Similarly, the sparsity of SVGP-VAE is determined by the number of inducing points, with a larger set generally leading to a richer approximation. In an ablation study, we analyze the effect of varying the number of inducing points in SVGP-VAE using the rotating MNIST experiments. We observe an improvement in Test MSE as the number of inducing points increases, leading to a consistent, albeit gradual, enhancement.
>
> A similar trend is observed with sLMM-VAE (Figure 4, 2nd plot), where test performance improves and stabilizes as more basis functions are added. These results indicate alignment in trends across the different GP approximation methods. While SVGP-VAE and sLMM-VAE deliver comparable and competitive results, the GSNN variants of LMM-VAE outperform both, achieving the strongest test performance in the rotating MNIST experiments.
>
>  |Model |Number of inducing points| Test MSE|
> |-------|-------|---------|
> |        | 16  |  $0.019 \pm 0.0005$       |
> |SVGP-VAE| 32  | $0.018 \pm 0.0002$       |
> |        | 48  |  $0.017 \pm 0.0003$      |
> |        | 64  |    $0.017 \pm 0.0007$       |
>
> ---
> [1] We assume that the discussion revolves around the SVGP-VAE baseline in our work by Jazbec et al., not the similarly named unpublished SGP-VAE by Ashman et al. (2021).

---

> > ### Author Response · Authors · 2025-03-12
> > **Official Comment by Authors (continuation)**
> >
> > >  explain 2) why the kernelised LMM-VAE remains robust against the curse of dimensionality when the number of covariates is high
> >
> > Our matrix $A$ has size d_hidden x M where M is the number of basis functions and d_hidden is the latent dimension. Notably, our setup does not rely on distances, unlike a GP with a stationary kernel, which encounters the usual challenges associated with measuring distances in high-dimensional spaces.
> >
> > The computational overhead associated with optimizing the LMM prior, driven by stochastic gradient descent (SGD), is $O(bF)$, where $F$ represents the total number of parameters and b represents the minibatch size. While the number of parameters needed for effective modelling increases with complexity of the problem and the number of covariates, this growth remains linear. This linear scalability makes LMM-VAE more efficient compared to methods that depend on inducing points.

---

> > > ### Comment · Reviewer_hND8 · 2025-03-12
> > > **Response to Further SVGP-VAE Comparison**
> > >
> > > Many thanks for providing these extra discussions. I am very satisfied by them and would be pleased to see the ablation study regarding the number of inducing points included somewhere in the final version of the paper. Regarding the robustness of the LMM-VAE against the curse of dimensionality, thanks for explaining this further to me---it seems my initial understanding was not exactly right.
> > >
> > > I hope to see the paper get accepted!

---

### Decision · Action_Editor_BiJR · 2025-04-24

**Recommendation:** Accept with minor revision

**Comment:**

Reviewer hND8 particularly appreciated the clarity of the final version, ultimately recommending acceptance. Reviewer ZB6M valued the idea but suggested the novelty is perhaps incremental when viewed from the lens of probabilistic PCA and GP-LVMs. Reviewer xWFG found the contribution valuable but requested further evaluation to situate LMM-VAE within the landscape of classical LMMs. (See above for details)

**Audience:**

The paper presents a well-argued and timely contribution, offering a complementary framework for modeling longitudinal data with complex covariate structures. TMLR explicitly welcomes exploratory contributions that lay the groundwork for future impact, and this submission aligns with that vision.

**Claims And Evidence:**

The papers propose **LMM-VAE**, a novel model that integrates linear mixed models (LMMs) with variational autoencoders (VAEs) to better handle high-dimensional longitudinal data. This should help address the limitations of GP-based VAEs, particularly their high computational cost and difficulty in modeling rich covariates. The LMM-VAE model is designed to offer better scalability and is applicable to both regression and classification tasks. The paper reports experiments on synthetic and real-world datasets, where the approach shows competitive or superior performance.

I thank the reviewers for the helpful and rigorous reviews. The reviewers see value in the method but, during the review, also raised questions about clarity, differences to other methods, etc.Overall, I am pleased with the response and think that the paper has meaningfully improved as a result.

One remaining concern is the lack of a thorough empirical comparison between LMM-VAE and classical LMMs, especially in scenarios where LMMs would typically perform well (e.g., low-dimensional, nearly-linear settings with repeated measures). While the authors position LMM-VAE as suited for more complex data, reviewer xWFG argues that demonstrating basic alignment with LMMs in simple settings would serve as a critical "sanity check" and help define the model's scope more precisely.

After careful discussion with reviewer xWFG, we would ask for the following as part of a minor revision:

> to what extent the data in-efficiency of LMM-VAEs makes them suitable / unsuitable for the kinds of tasks that LMMs do better in, given that they should, in principle, be able to model overlapping data generating processes.

> This could be done via a simulation study and/or real data, where the true effects/parameters of interest are already known. If the authors would argue that parameter estimation is not the goal of LMMs (which I would disagree with), then at the very least the comparison should be made on a predictive basis (i.e. compare the o.o.s. MSE / predictive metric between the two models on a dataset which can be modelled more/less/completely linearly) <as a function of sample size>.

This should be a kind of 'sanity check' that LMM-VAEs can achieve the same as LMMs can as the data-size increases. I concur that such a simulation study is a nice add-on and the missing piece for acceptance.

---

> ### Author Response · Authors · 2025-05-24
> **Revised manuscript**
>
> We thank the Action Editor and the Reviewers for their constructive feedback. In response, we have uploaded a revised version of the manuscript and included the additional experiments, as requested, in Appendix E.2 as part of the revision.